# Glycine acylation and trafficking of a new class of bacterial lipoprotein by a composite secretion system

Christopher Icke[1], Freya J Hodges[1], Karthik Pullela[1], Samantha A McKeand[2], Jack Alfred Bryant[2], Adam F Cunningham[2,3], Jeff A Cole[2], Ian R Henderson[1]*

[1]Institute for Molecular Bioscience, University of Queensland, Brisbane, Australia; [2]Institute of Microbiology and Infection, Birmingham, United Kingdom; [3]Institute of Immunology and Immunotherapy, University of Birmingham, Birmingham, United Kingdom

**Abstract** Protein acylation is critical for many cellular functions across all domains of life. In bacteria, lipoproteins have important roles in virulence and are targets for the development of antimicrobials and vaccines. Bacterial lipoproteins are secreted from the cytosol via the Sec pathway and acylated on an N-terminal cysteine residue through the action of three enzymes. In Gram-negative bacteria, the Lol pathway transports lipoproteins to the outer membrane. Here, we demonstrate that the Aat secretion system is a composite system sharing similarity with elements of a type I secretion systems and the Lol pathway. During secretion, the AatD subunit acylates the substrate CexE on a highly conserved N-terminal glycine residue. Mutations disrupting glycine acylation interfere with membrane incorporation and trafficking. Our data reveal CexE as the first member of a new class of glycine-acylated lipoprotein, while Aat represents a new secretion system that displays the substrate lipoprotein on the cell surface.

**\*For correspondence:**
i.henderson@imb.uq.edu.au

**Competing interests:** The authors declare that no competing interests exist.

## Introduction

Protein acylation by the covalent attachment of fatty acids occurs for hundreds of proteins in eukaryotic and prokaryotic organisms. This event confers distinct biochemical properties, enabling acylation to regulate intracellular trafficking, subcellular localisation, protein-protein and protein-lipid interactions, and are of obvious importance to cell biology. As a consequence, lipoproteins are key components of bacterial pathogens and have been targeted for antibiotic and vaccine development (*Dev et al., 1985*). In Gram-negative bacteria, lipoproteins are transported across the inner membrane from the cytosol to the periplasm by the Sec pathway. However, in Gram-positive cells, lipoproteins can also be secreted by the TAT pathway (*Thompson et al., 2010*; *Widdick et al., 2011*). The signal peptide inserts into the inner membrane and a diacylglycerol is attached by Lgt to the sulphur moiety of the invariant cysteine at the +1 position of the lipoprotein. The signal sequence is then cleaved by Lsp (signal peptidase II) exposing the N-terminal amine group of the cysteine for monoacylation by Lnt (*Gupta et al., 1993*; *Mizushima, 1984*). The mature triacylated lipoprotein remains embedded in the inner membrane or is localised to the inner leaflet of the outer membrane by the essential Lol pathway. LolCE, in combination with the ATPase LolD, extracts the triacylated lipoprotein from the inner membrane for transport across the periplasm by LolA, where it is then incorporated into the inner leaflet of the outer membrane by LolB. While the majority remain periplasmically located, in recent years, there have been a number of descriptions of surface localised outer membrane lipoproteins (*Baldi et al., 2012*; *Cowles et al., 2011*; *Konovalova et al., 2014*). However, the mechanism of translocation across the outer membrane to the cell surface remains poorly understood.

Previously, we described two outer membrane proteins (OMPs), CexE and Aap, associated with enterotoxigenic (ETEC) and enteroaggregative *Escherichia coli* (EAEC), respectively (*Crossman et al., 2010*; *Sheikh et al., 2002*). CexE has been implicated in prolonged bacterial shedding and increased severity of infection (*Rivas et al., 2020*), whereas Aap influences biofilm formation and gut colonisation (*Belmont-Monroy et al., 2020*; *Sheikh et al., 2002*). Despite Aap and CexE sharing only 18% amino acid identity, both proteins are secreted by the Aat system (*Belmont-Monroy et al., 2020*; *Nishi et al., 2003*; *Rivas et al., 2020*). The Aat system requires five proteins (AatPABCD) to facilitate protein secretion, two of which bear resemblance to components of the type I secretion system (T1SS): an OMP and a periplasmic adaptor protein (PAP). In contrast to the T1SS, the CexE and Aap substrate molecules are translocated using a two-step mechanism. First, Aap/CexE is translocated across the inner membrane into the periplasm by the Sec pathway (*Pilonieta et al., 2007*). It then enters the Aat system to be secreted across the outer membrane (*Belmont-Monroy et al., 2020*; *Nishi et al., 2003*; *Rivas et al., 2020*).

While further characterising the two-step secretion mechanism of the Aat system, we noticed that during secretion, CexE is post-translationally modified by the Aat system and this modification is required for the secretion. Here, we reveal that following cleavage of the Sec-dependent signal sequence, the N-terminus of CexE is modified by the addition of an acyl chain. We demonstrate that AatD is a homolog of the apolipoprotein *N*-acyltransferase (Lnt). We reveal that AatD is both necessary and sufficient for monoacylation of CexE and Aap. However, in contrast to Lol lipoprotein substrates, CexE lacks an N-terminal cysteine and instead an invariant glycine is the site of acylation. Furthermore, we demonstrate that the addition of an N-terminal glycine to the coding sequence of a heterologous protein was sufficient for this novel AatD-catalysed acylation event. We propose that Aap and CexE are members of a novel class of lipoprotein that are secreted through the Aat system, which is a conglomeration of the Lol pathway and a T1SS. We reveal that AatD is a new acyltransferase with glycine as the target of N-palmitoylation. Consequently, we reveal a new function for acylation-protein secretion.

## Results

### Distribution of the Aat system

The Aat system was first identified in EAEC, where it corresponds to the molecular probe (CVD432) used to define this *E. coli* pathovar (*Baudry et al., 1990*; *Nishi et al., 2003*). In order to determine whether the Aat system was more widespread, each of the Aat proteins from ETEC H0407 was used to search the non-redundant protein sequence database using repetitive iterations of the PSI-BLAST algorithm. The full Aat system and an Aap or CexE homolog was identified in 826 separate nucleotide accessions (*Supplementary file 1*). This revealed that the Aat system is distributed more widely than initially anticipated and is encoded in pathogens with diverse mechanisms of virulence such as ETEC, EAEC, enteropathogenic *E. coli*, Shiga-toxin producing *E. coli*, *Shigella* sp., *Salmonella enterica*, *Citrobacter rodentium*, *Providencia alcalifaciens*, and *Yersinia entercolitica* (*Crossman et al., 2010*; *Petty et al., 2010*; *Rivas et al., 2020*).

To understand the conservation of the Aat system, the organisation of the *aat* operon was examined for each of the 826 genomes identified above. The relative genomic distance between each of the *aat* genes and *aap/cexE* was calculated from their boundaries on the nucleotide accession. From this analysis six separate organisations of the *aat* operon were identified (*Figure 1*). In the most common organisation, *aap/cexE* are separated from the *aatPABCD* operon by at least 1 kb. This accounted for just over half of the *aat* systems identified. In just under a quarter of *aat* operons identified, *aap/cexE* are within 1 kb of the *aatPABCD* operon either within 400 bp upstream (18.4%) or 1 kb downstream (4.8%). In 25% of the *aat* systems identified, *aatD* was separated from *aatPABC* by >1 kb. The most common of these organisations was *aatD* separated from the other members of the *aat* system and substrate protein (11.9%), closely followed by *aatD* and *aap/cexE* together but greater than 1 kb distance from other *aat* genes (9.0%). Finally, in the least common example, *aatD*, *aap/cexE,* and *aatPABC* were all separated by greater than 1 kb (4.1%). We did not identify any examples of *aatD* encoded between 39 bp and 1 kb from the stop codon of *aatC*. However, despite these differences, in every example, the *aatPABC* genes formed a single operonic unit located with

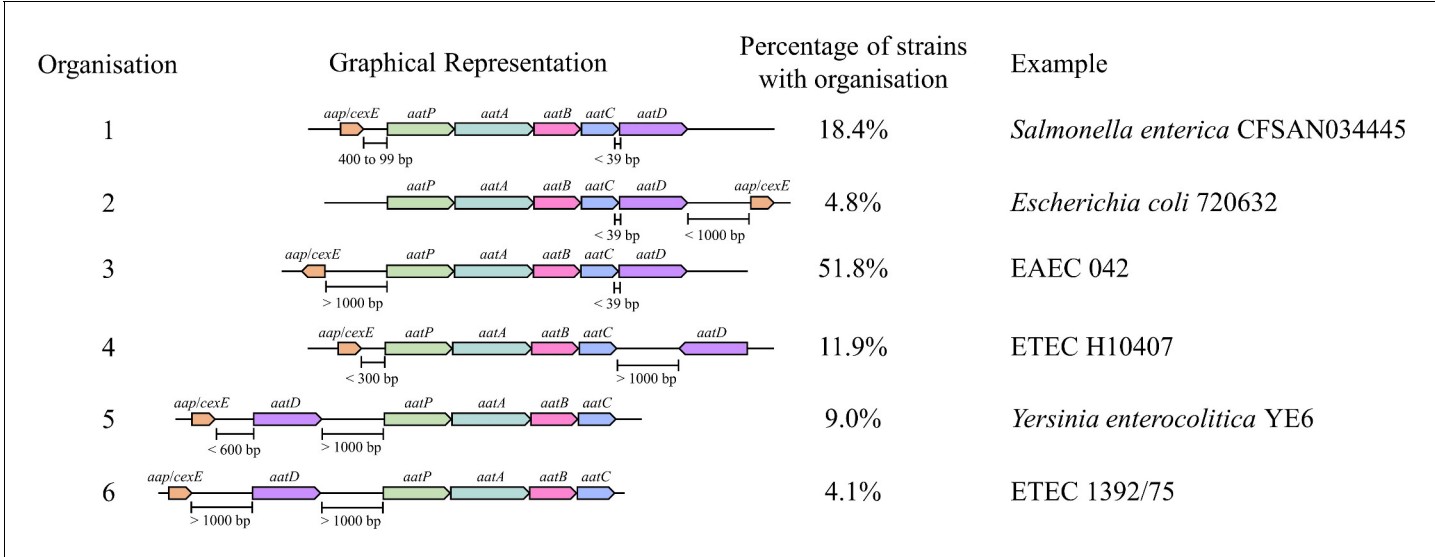

**Figure 1.** Organisation of the Aat operon. Unique Aat amino acid sequences were detected using PSI-BLAST. These sequences were used to identify the strains that encoded them in the NCBI identical protein groups. As the *aat* genes were present on contigs of whole genome sequencing projects, it was not possible to assess if a strain encoded a gene on the chromosome or plasmid. Instead, contigs were used to identify the complete Aat system. This analysis does not include strains that might encode the *aat* genes or *aap/cexE* on a separate genomic element. However, a total of 827 complete Aat systems were identified in the same nucleotide accession. The positions of these genes were used to assess the organisation of the *aat* operon. From this assessment five different classes of *aat* operon organisation were defined.

The online version of this article includes the following source data for figure 1:

**Source data 1.** *aatA* gene positions.
**Source data 2.** *aatB* gene positions.
**Source data 3.** *aatC* gene positions.
**Source data 4.** *aatD* gene positions.
**Source data 5.** *aatP* gene positions.
**Source data 6.** *aap* or *cexE* gene positions.
**Source data 7.** Calculated distance between genes.
**Source data 8.** Distances and designations.

*aatD* and *aap/cexE* on a chromosomal pathogenicity island or a large virulence plasmid. These data suggest that AatPABCD have a role in Aap/CexE secretion.

## Aat-dependent secretion of CexE

To test whether all of the *aat* genes are required for CexE secretion, we constructed single gene deletion mutants of *cexE* and each *aat* gene in ETEC H10407 pCfaD. The pCfaD plasmid encodes the CfaD transcriptional activator under the control of an arabinose inducible promoter allowing constitutive expression of the CfaD-dependent *cexE* and the *aat* genes in the presence of arabinose (*Hodson et al., 2017*; *Pilonieta et al., 2007*). SDS-PAGE analysis of culture supernatant fractions collected from ETEC H10407 pCfaD revealed a protein with an apparent molecular mass of 11.8 kDa that could be detected by western blotting with CexE-specific polyclonal antibodies (*Figure 2A*). In contradistinction to the parent strain, no protein was detected in the culture supernatant fractions derived from cultures of the ETEC H10407 pCfaD *aat* or *cexE* mutants (*Figure 2A*). However, to ensure that the lack of CexE in the culture supernatant was not due to the lack of CexE production, but rather a result of an inability to secrete CexE, the presence of CexE in whole-cell lysates was determined (*Figure 2B*). CexE was detected by western blotting with CexE-specific antibodies in whole-cell lysates of ETEC H10407 pCfaD and all *aat* mutants. In contrast, CexE could not be detected in the *cexE* null mutant (*Figure 2*). From these data we conclude that the Aat system is essential for CexE secretion.

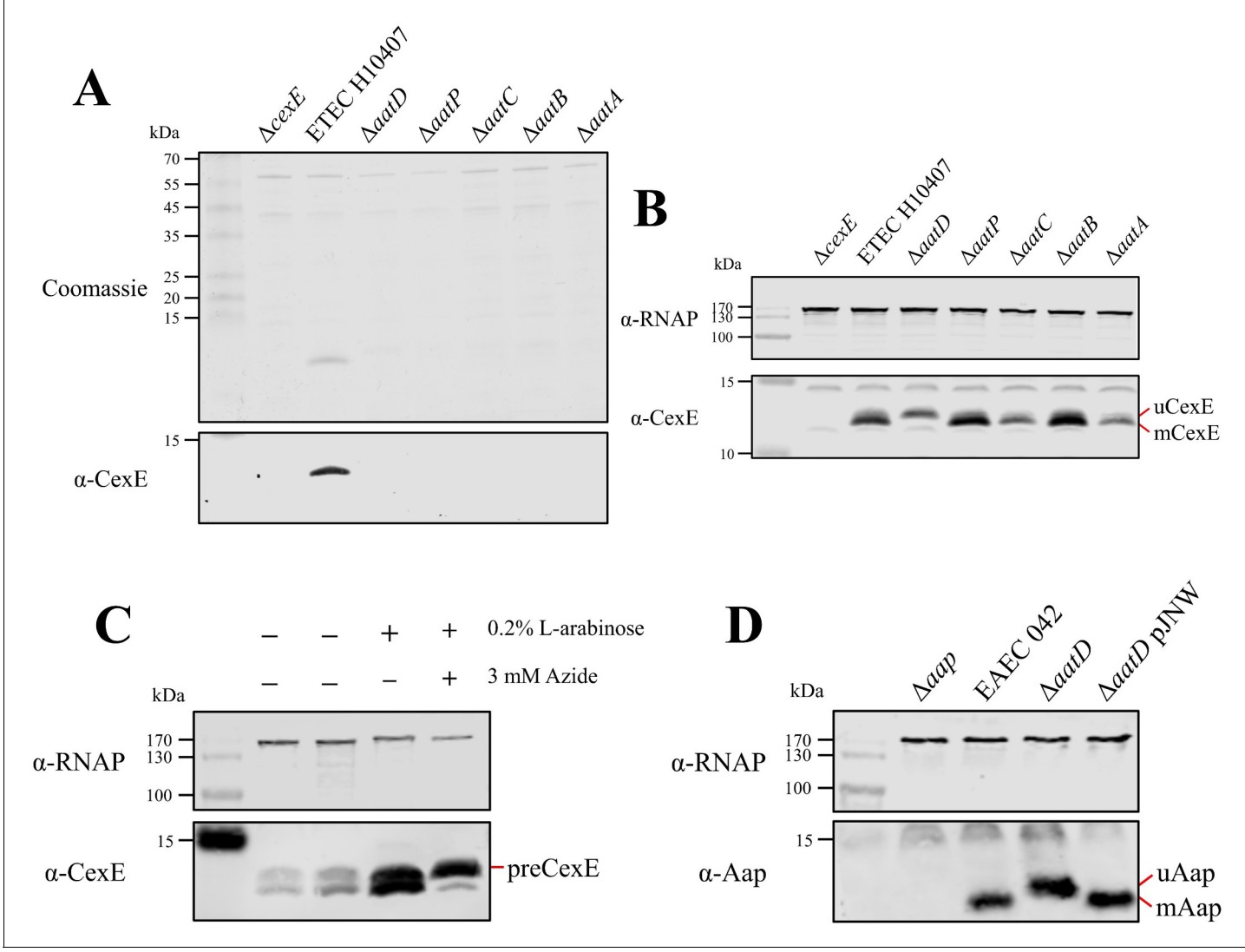

**Figure 2.** AatD post-translational modification of Aap and CexE. (**A**) Culture supernatant of ETEC H10407 and *aat* mutants harbouring pCfaD grown in lysogeny broth (LB) supplemented with L-arabinose. Cells were removed from the culture supernatant and remaining protein precipitated with trichloroacetic acid. Protein samples were separated by Tris-tricine SDS-PAGE and stained with Coomassie or transferred to nitrocellulose for western blotting with polyclonal antibodies against CexE. (**B**) Whole-cell lysates of the *aat* mutants and parental strain separated by Tris-tricine SDS-PAGE. (**C**) Whole-cell lysates of ETEC H10407 grown in LB with or without azide. (**D**) Whole-cell lysates of EAEC 042, *aap*, *aatD*, and *aatD* complemented strains grown in DMEM (Dulbecco's Modified Eagle Medium) high glucose. The positions of uCexE, mCexE, uAap, mAap, and preCexE are indicated as appropriate. CexE and Aap were detected by western blotting (α-CexE and α-Aap, respectively) and RNA polymerase (α-RNAP) was included as a loading control where appropriate.

The online version of this article includes the following source data and figure supplement(s) for figure 2:

**Source data 1.** Western blot and Coomassie stained gels of culture supernatant fractions.
**Source data 2.** Western blot of whole-cell lysates derived from wild-type and mutant strains.
**Source data 3.** Western blot of whole-cell lysates after growth in sodium azide.
**Source data 4.** Western blot of whole-cell lysates showing Aap modification in the presence and absence of *aatD*.
**Figure supplement 1.** Estimated molecular mass of uCexE and mCexE.

## Post-translational modification of Aat substrate molecules

Unexpectedly, when the whole-cell lysates of ETEC H10407 pCfaD and its isogenic *aat* mutants were examined by western blotting, CexE appeared as two bands in the parental strain and each of the *aat* mutants except for *aatD* (*Figure 2B*). The apparent molecular mass of the two CexE subtypes was determined from the migration of the different protein species on an SDS-PAGE gel relative to

the molecular weight ladder (*Figure 2—figure supplement 1*). The upper band had an apparent molecular mass of 12.7 kDa (uCexE), while the lower band (mCexE) had an apparent molecular mass of 11.8 kDa, which is consistent with the secreted form of CexE recovered from the supernatant fractions. As CexE is predicted to be exported into the periplasm by the Sec pathway (*Pilonieta et al., 2007*), the inefficient cleavage of a signal peptide could be responsible for the difference in the molecular mass of uCexE compared to mCexE. To investigate this, ETEC H10407 pCfaD was grown in the presence of sodium azide. Sodium azide inhibits SecA to prevent translocation of proteins across the inner membrane and subsequent cleavage of the signal peptide (*Huie and Silhavy, 1995*; *Oliver et al., 1990*). In the absence of sodium azide, the mCexE and uCexE forms could be detected in whole-cell lysates as observed previously (*Figure 2B and C*). However, in the presence of sodium azide, a protein (preCexE) with an apparent molecular mass of 13.2 kDa was detected (*Figure 2C*). These observations suggest that the difference in molecular mass between uCexE and mCexE was not due to inefficient cleavage of the signal peptide but was due to a post-translational event mediated by AatD.

A size change in the CexE homolog Aap has not been reported before despite numerous investigations of the Aat system. To determine whether the migration of Aap was altered by AatD, the effect of an *aatD* deletion on the apparent size of Aap was investigated in EAEC 042. EAEC 042, an *aap* mutant, an *aatD* mutant, and an *aatD* mutant complemented with pJNW were grown in DMEM-high glucose (HG); the pJNW plasmid encodes the complete *aat* operon from EAEC 042. Whole-cell lysates of each strain were analysed by SDS-PAGE. As noted above for CexE, the deletion of *aatD* in EAEC 042 resulted in an apparent increase in size of Aap (*Figure 2D*). The size change of Aap in the *aatD* mutant could be restored to the wild-type size by complementation with the pJNW plasmid. From these data we conclude that AatD can modify Aap and CexE post-translationally and that this modification is required for secretion of the mature substrate molecule to the extracellular milieu.

## Predicted functions of Aat components

To assist our understanding of the contribution of each Aat component to the secretion of CexE, we constructed hidden Markov models (HMM) for individual Aat proteins. These were used to search the Uniprot database for distant homologs. In agreement with previous publications, AatA is a homolog of the trimeric OMP TolC (*Figure 3—figure supplement 1*) and AatB is a PAP (*Figure 3—figure supplement 2*), components associated with T1SS and drug efflux pumps (*Nishi et al., 2003*). In contrast, AatD was not homologous to any T1SS or efflux-associated proteins. Instead, AatD was homologous to apolipoprotein *N*-acyltransferase (Lnt), which is a member of the carbon-nitrogen hydrolase (C-N hydrolase) family (PF00795) required for bacterial lipoprotein acylation and transport via the Lol system (*Figure 3A*). Comparison of the C-N hydrolase domain of AatD with other members of PF00795 revealed that AatD forms a clade with Lnt suggesting that they are functionally related. The ATPase domain of AatC is part of a clade containing the ATPase domains of LolD, the ATPase subunit of the LolCDE lipoprotein transporter (*Figure 3B*). Similarly, AatP is more closely related to LolC and LolE than typical ABC transporters of the T1SS, such as HlyB (*Figure 3C*). The homology of AatP, AatC, and AatD to proteins of the Lol pathway suggests that the Aat system is a composite system of a T1SS and the Lol lipoprotein trafficking system.

## AatD-mediated acylation of mCexE

As AatD is a homolog of the Lnt acyltransferase, we hypothesised that mCexE represents an acylated form of CexE. To test this hypothesis, we used 17-ODYA, an 18-carbon alkyne fatty acid that can be conjugated to an azide linked fluorescent molecule by a copper(I)-catalysed azide-alkyne cycloaddition (CuAAC) reaction to investigate CexE lipidation. The *cexE* gene was introduced into a *cexE* mutant or a *cexE aatD* double mutant on a plasmid termed pACYC-*cexE*-6His, which encodes *cexE* under the transcriptional control of its native promoter. This allowed the production of a C-terminally His-tagged variant of CexE. Both strains harbouring pACYC-*cexE*-6His were grown in the presence or absence of 17-ODYA. The bacterial lipoprotein YraP was included as a positive control. CexE and YraP were isolated from each culture by virtue of their C-terminal 6xHis-tag by using nickel affinity chromatography. The purified proteins were then linked to an azide derivative of Alexa Fluor 488 using a CuAAC reaction. After the CuAAC reaction, the CexE and YraP samples were separated by SDS-PAGE and fluorescence was measured (*Figure 4*). In the presence of AatD and 17-ODYA,

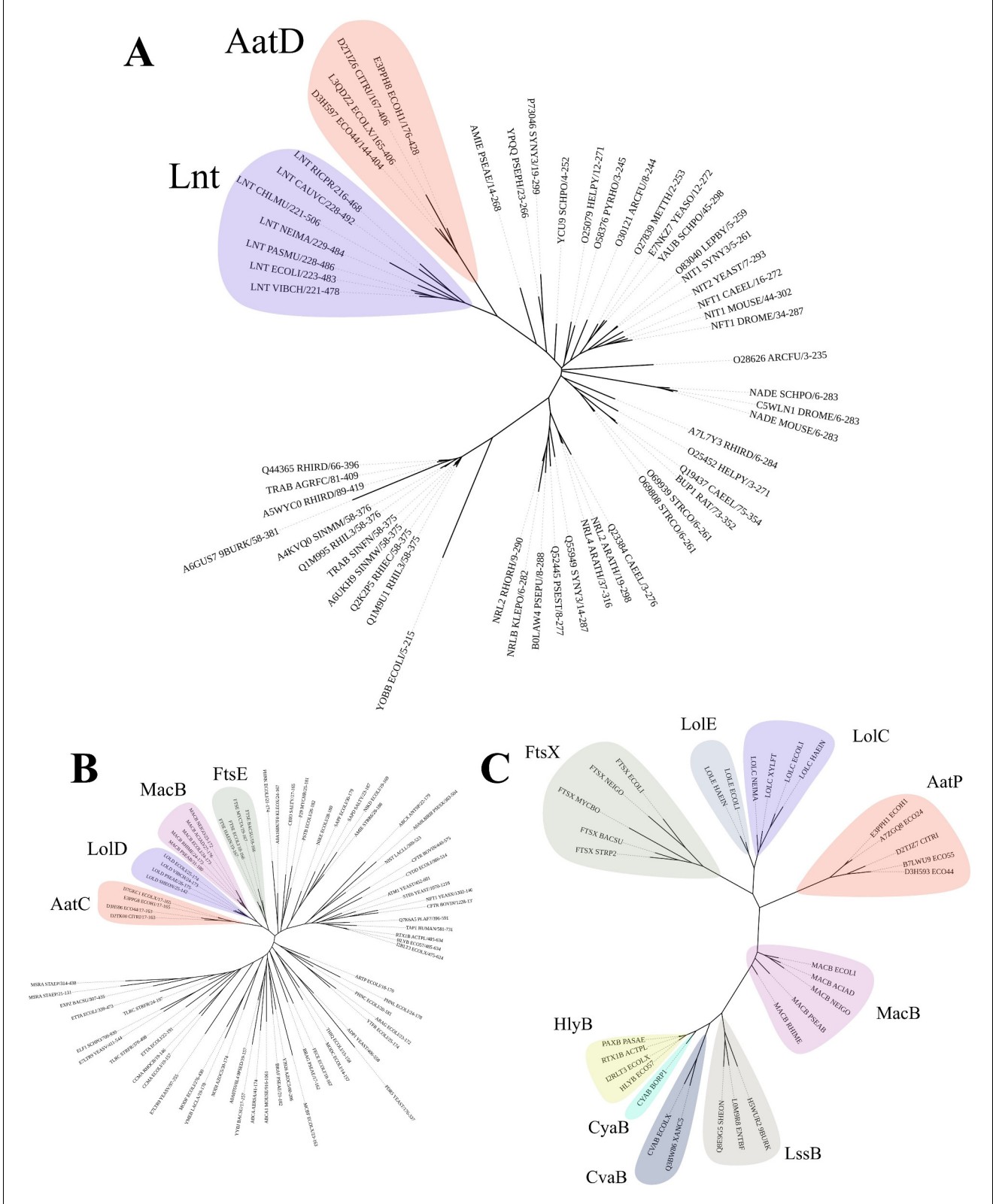

**Figure 3.** Phylogenetic analysis of AatD, AatP, and AatC. Sequences were aligned using Clustal omega and the tree was calculated using RAxML. (**A**) The carbon-nitrogen(C-N) hydrolase domains of C-N hydrolase family seed sequences were aligned to the C-N hydrolase domains of AatD sequences from ETEC H10407, EAEC 042, *Citrobacter rodentium* ICC168, and *Escherichia coli* KTE75. (**B**) Phylogram of the ATPase domains of the pfam ABC

*Figure 3 continued on next page*

*Figure 3 continued*

transporters (PF00005) and AatC sequences. (C) Phylogram of type I secretion system (T1SS) ABC transporters, Lol ABC transporters, and AatP sequences.

The online version of this article includes the following source data and figure supplement(s) for figure 3:

**Source data 1.** AatD and carbon-nitrogen (C-N) hydrolase Pfam family sequences used for tree.
**Source data 2.** Sequences used to generate AatP tree.
**Figure supplement 1.** Phylogram of AatA, TolC, and other outer membrane protein (OMP) associated with T1SS and efflux pumps.
**Figure supplement 1—source data 1.** AatA and outer membrane protein (OMP) sequences used for tree.
**Figure supplement 2.** Phylogram of AatB and periplasmic adaptor proteins (PAPs) involved in type I secretion system (T1SS) and efflux pumps.
**Figure supplement 2—source data 1.** AatB and periplasmic adaptor protein (PAP) sequences used for tree.

CexE was fluorescently labelled; in the absence of either 17-ODYA or AatD, no fluorescence was observed. Moreover, only mCexE was labelled indicating that uCexE is an unacylated form of CexE; counterintuitively, the acylated form (mCexE) migrates faster on SDS-PAGE than the unacylated uCexE. Furthermore, only the acylated versions of Aap and CexE were present in the culture supernatant (*Figure 4—figure supplement 1*). As acylated proteins associate with membranes, we examined the cellular compartmentalisation of mCexE. ETEC H10407 pCfaD and isogenic *aat* mutants were grown in lysogeny broth (LB) supplemented with L-arabinose. Subsequently, cells were collected by centrifugation, lysed and the total membrane fraction was harvested by centrifugation. After separation by SDS-PAGE, only mCexE was detectable by western blotting in the membrane

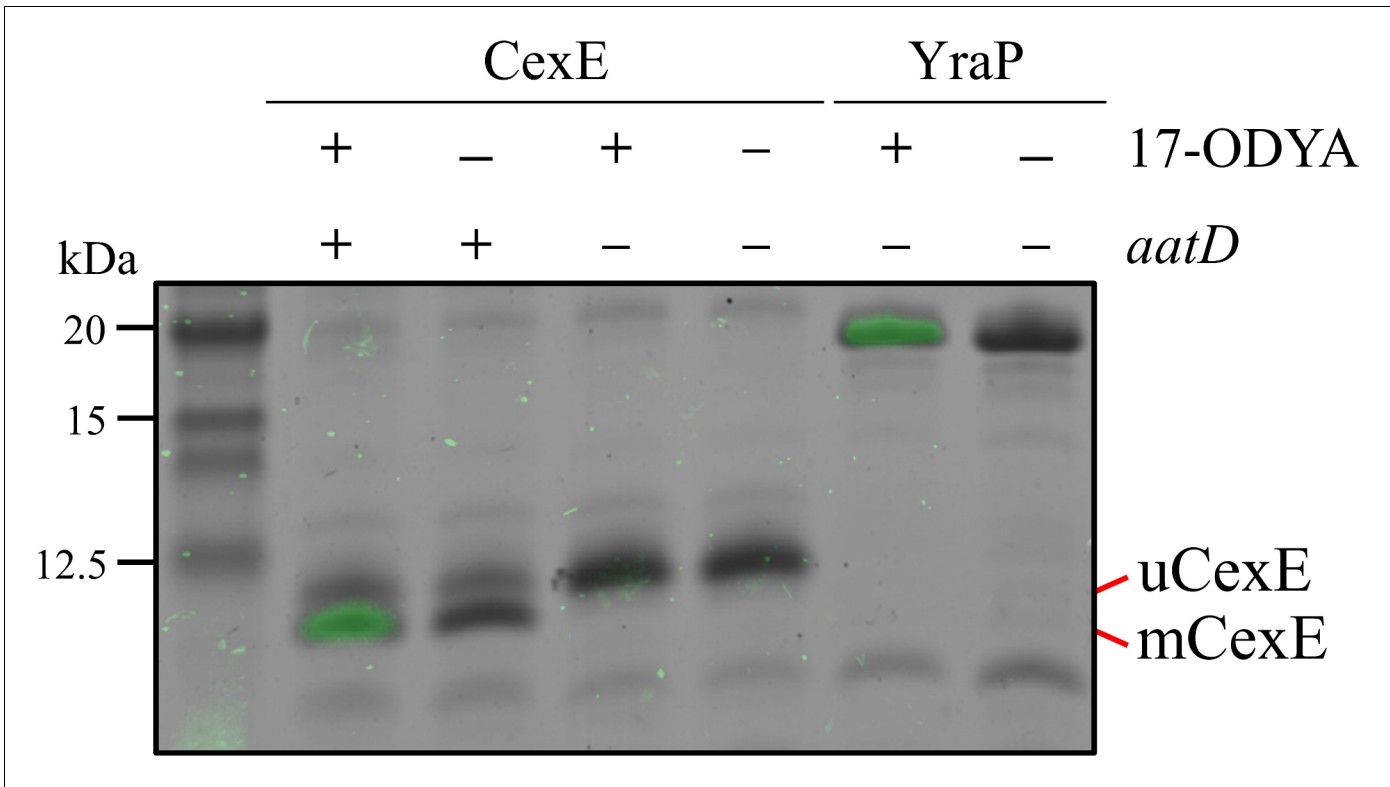

**Figure 4.** Incorporation of 17-ODYA into CexE in the presence of AatD. His-tagged CexE was isolated from *cexE* or *cexE aatD* double mutants harbouring pACYC-*cexE*-6His grown in the presence or absence of 17-ODYA and separated by SDS-PAGE. An azide linked Alexa Fluor 488 was conjugated to the alkyne moiety present in 17-ODYA by CuAAC. The incorporation of 17-ODYA into the target protein was detected by fluorescence (green bands) and the image was overlaid on the image of the SDS-PAGE gel. A known lipoprotein YraP was used as a positive control.

The online version of this article includes the following figure supplement(s) for figure 4:

**Figure supplement 1.** 17-ODYA labelling of secreted Aap and CexE.
**Figure supplement 2.** Membrane localisation of CexE.

fraction of the parent strain and the *aatPABC* mutants; no mCexE could be detected in the membranes recovered from the *aatD* mutant (**Figure 4—figure supplement 2**). In contrast, both mCexE and uCexE could be detected in whole-cell lysates of ETEC H10407 pCfaD and only uCexE could be detected in whole-cell lysates of the *aatD* mutant. These data suggest that AatD is an acyltransferase that mediates post-translational modification of CexE by the addition of one or more acyl chains.

To further determine whether AatD was necessary to mediate the acylation of mCexE, plasmids encoding AatD and CexE were introduced into the laboratory strain *E. coli* BL21(DE3), which does not encode any of the Aat proteins or Aap/CexE. *E. coli* BL21 was transformed with pET26b-*cexE* and pACYC-*aatD* or their respective empty vector controls pET26b and pACYCDuet-1. These strains were grown in LB and the production of AatD and CexE was induced by IPTG. The production of CexE was monitored via western blotting (**Figure 5**). CexE was not detected in strains that contained pET26b. uCexE was produced in the absence of AatD at the size expected for the unmodified form after cleavage of the signal sequence. However, when CexE and AatD were produced together, mCexE migrated further (**Figure 5**). These data suggest AatD was necessary for the modification of CexE.

## Catalytic residues of AatD

The C-N hydrolase family of proteins have a known conserved catalytic triad, which in the case of Lnt is E267, K335, and C387. Another Lnt residue E343 helps to stabilise the catalytic site (**Vidal-Ingigliardi et al., 2007**). Mutation of any of these residues to alanine results in a loss of Lnt function (**Gélis-Jeanvoine et al., 2015**). To assess the conservation of the catalytic residues, the AatD sequence of EAEC 042 and ETEC H10407 and Lnt from *E. coli* MG1655 were aligned using Clustal Omega. Each of the residues associated with the catalytic site was conserved in both versions of AatD (**Figure 6A**). The C-N hydrolase catalytic triad is conserved in AatD: E217, K278, and C325. Also, Lnt E343 is conserved at position E286 in AatD. Further AatD sequences were identified using

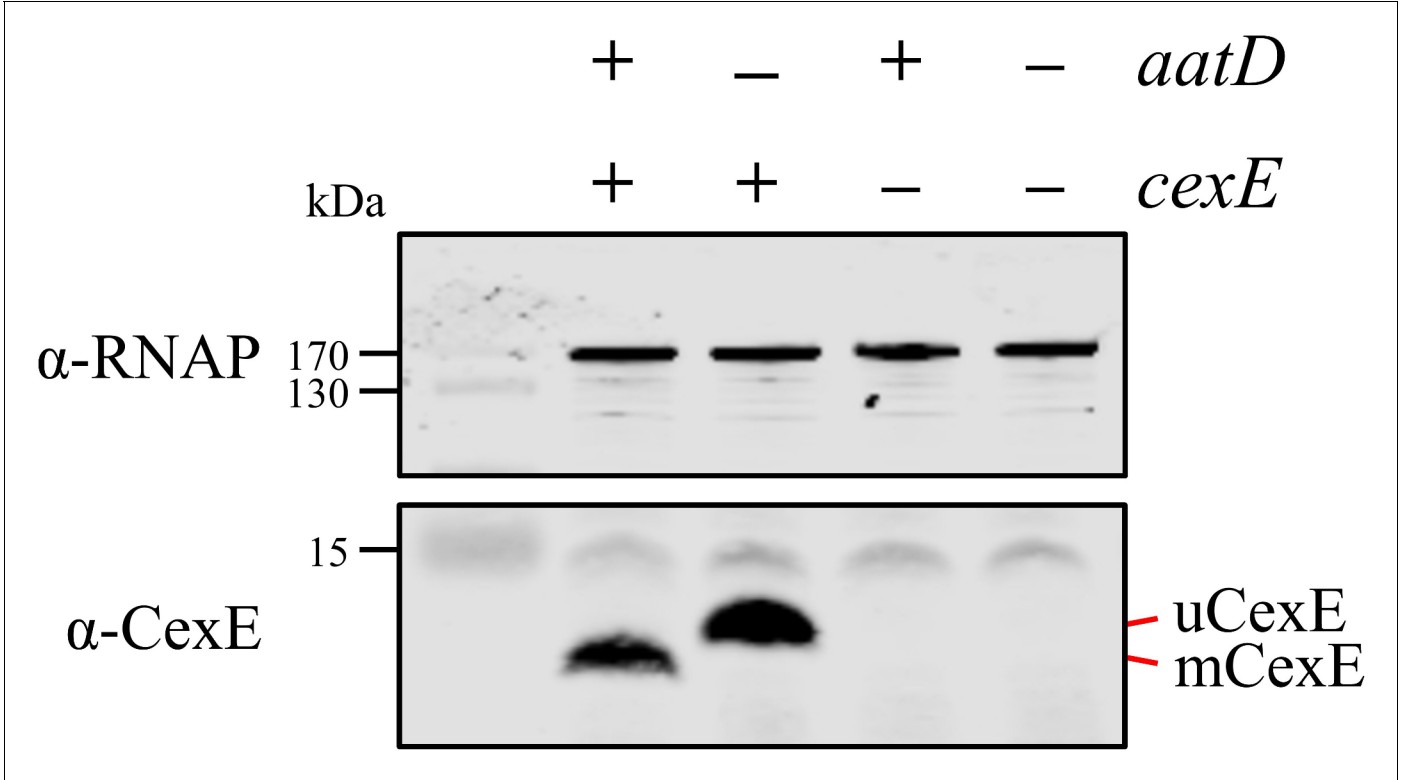

**Figure 5.** Recapitulation of CexE acylation in *E. coli* BL21(DE3). *E. coli* BL21(DE3) was transformed with pET26b, or pET26b-*cexE* and pACYCDuet-1 or pACYC-*aatD*. Cultures were grown in lysogeny broth (LB) and protein production was stimulated with isopropyl β-D-1-thiogalactopyranoside (IPTG). Whole-cell lysate samples were taken and separated by Tris-tricine SDS-PAGE. CexE was detected using a polyclonal antibody. RNAP was used as a loading control. The position of mCexE and uCexE is indicated.

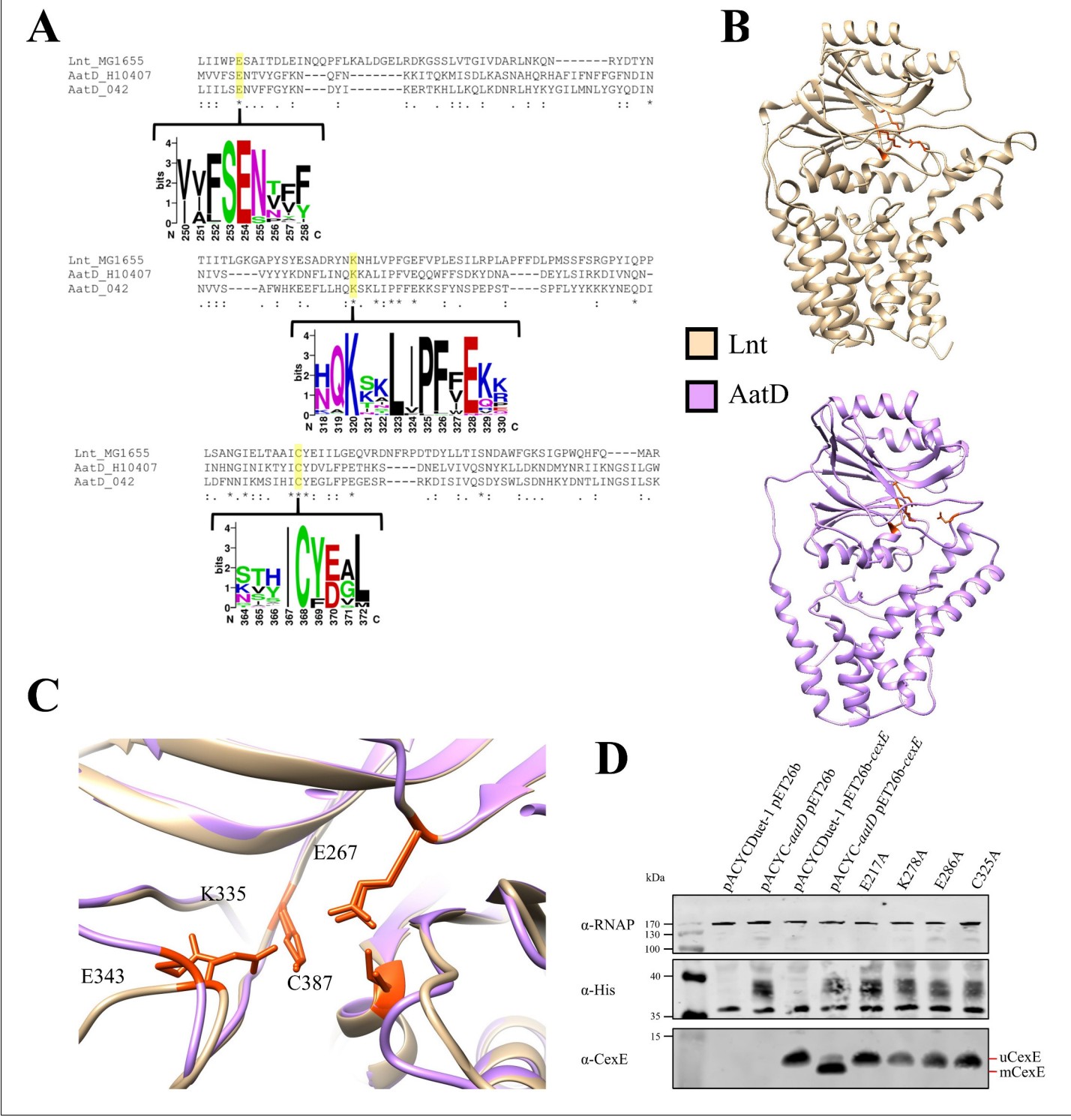

**Figure 6.** Catalytic residues of AatD. (**A**) AatD from EAEC 042 and ETEC H10407 were aligned with Lnt from *Escherichia coli* MG1655. AatD sequences identified by PSI-BLAST were aligned and used to create a WebLogo. The catalytic residues of the carbon-nitrogen (C-N) hydrolase family are highlighted. (**B**) The structure of Lnt compared to the predicted structure of ETEC H10407 AatD. The catalytic residues of Lnt and AatD are highlighted in orange. (**C**) Magnified view of the catalytic site of Lnt with the predicted structure of AatD superimposed. Residues are numbered as they appear in Lnt. (**D**) WCL of *E. coli* BL21(DE3) expressing CexE in the presence of AatD and each of the four single site substitution derivatives of AatD. CexE was detected using an anti-CexE polyclonal antibody, AatD was detected with an anti-His tag antibody, and RNAP was used as a loading control.

PSI-BLAST. These sequences were aligned and a WebLogo was produced for the regions flanking each of the catalytic residues; the four residues were 100% conserved in all AatD sequences identified. In addition, when a structure of AatD predicted by Phyre2 (*Kelley et al., 2015*) was compared to Lnt (PDB: 5N6L) (*Wiktor et al., 2017*), the four AatD residues (E217, K278, E286, and C325) were superimposed on the Lnt catalytic triad suggesting that these residues were required for acylation of CexE (*Figure 6B and C*). To test this hypothesis, pACYC-*aatD* derivatives encoding mutant derivatives of AatD with alanine substitutions at the conserved E217, K276, E286, or C325 were introduced into *E. coli* BL21(DE3) pET26b-*cexE*. Expression of AatD and CexE was induced by the addition of isopropyl β-D-1-thiogalactopyranoside (IPTG) during growth in LB. AatD production was detected via a C-terminal 6 His tag (*Figure 6D*) and the level of production was similar for the wild-type and mutant derivatives. As expected, mCexE was only present in the strain encoding the wild-type *aatD* sequence, whereas only unmodified uCexE was detected in mutants harbouring the E217A, K276A, E286A, or C325A (*Figure 6D*). These observations suggest that the mechanism of protein acylation by AatD and Lnt is similar and that AatD is necessary for the acylation event.

## An N-terminal glycine is required for acylation

Due to the homology and functional similarity between AatD and Lnt, a likely site for acylation of CexE is the N-terminal amino acid immediately after the signal sequence. SignalP predictions suggested that CexE possesses a Sec-dependent signal sequence cleaved by signal peptidase I between alanine at position 19 and glycine at position 20 (*Figure 7A* and *Supplementary file 2*). N-terminal amino acid sequencing of uCexE revealed the amino acid sequence GGGNS confirming that glycine at position 20 formed the N-terminal amino acid of the processed protein. However, bioinformatic analyses of the amino acid sequences of 224 distinct CexE homologs failed to identify in their signal sequences the presence of a 'lipobox' (*Figure 7—figure supplement 1*); lipoboxes are required for recognition of lipoproteins by the Lol system and thus acylation by Lnt (*Babu et al., 2006*). Further analyses revealed limited sequence identity between the signal sequences of the 224 CexE homologs indicating that an alternative lipobox is not present. Moreover, neither CexE nor any other homolog possessed a cysteine residue adjacent to the signal sequence; for bacterial lipoproteins, the N-terminal cysteine of the mature protein is the target of Lnt mediated acylation. Since CexE does not contain features of a typical bacterial lipoprotein, it must be recognised and acylated by AatD in a manner different to the acylation of lipoproteins by Lnt.

Further scrutiny of the amino acid sequences of all CexE homologs revealed that for all CexE and Aap proteins, the first five amino acids immediately after the signal peptide cleavage site are a conserved mix of glycine and serine residues (*Figure 7B*). The presence of an invariant N-terminal glycine residue immediately downstream of the predicted signal sequence suggested that this might be the site of acylation. To test this hypothesis, we mutated each of the N-terminal residues of CexE to glutamic acid and observed the effect on acylation by western blotting. The amount of total fluorescence of the secondary antibody bound to primary polyclonal anti-CexE antibody was measured and compared for each mutant. The percentage of acylated mCexE was calculated for each mutant. The migration of the G20E mutant was similar to uCexE indicating that the change to glutamic acid completely abolished acylation (*Figure 7C*). In addition, the acylation of the G21E mutant was significantly reduced (p-value=0.0013). However, there was no significant reduction in acylation of the G22E, N23E, or S24E mutants (*Figure 7C*). Moreover, CexE was no longer secreted without the acylation at G20 (*Figure 7—figure supplement 2*). These data indicate that the acylation of CexE is highly dependent on the N-terminal glycine at position 20 and to a lesser extent on the second glycine at position 21.

## Mass spectrometric analysis of CexE acylation

Mutation of the conserved N-terminal glycine can abolish acylation of CexE, and Lnt acylates the N-terminal amine of the signal peptidase processed lipoprotein, therefore we hypothesised that AatD would acylate the N-terminal amine of G20. To test this, we first purified His-tagged mCexE and uCexE. mCexE was solubilised from the membrane fraction of ETEC H10407 pCfaD pACYC-*cexE*-6His using Triton X-100 while uCexE was purified from the soluble fraction of the cell lysate of ETEC H10407Δ*aatD* pCfaD pACYC-*cexE*-6His. Both mCexE and uCexE were then purified by nickel affinity chromatography. Both forms of CexE were subjected to trypsin digestion followed by LC-

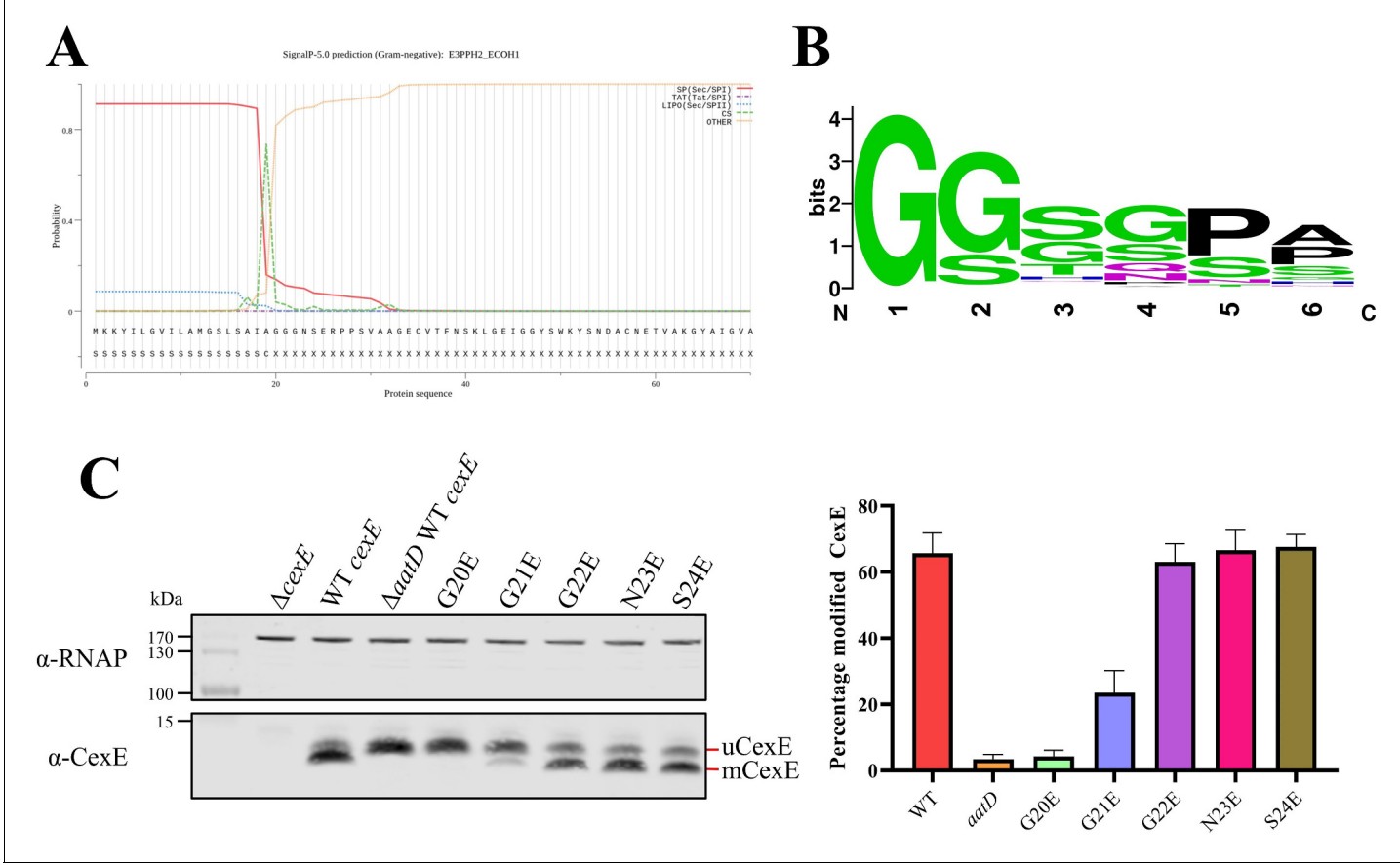

**Figure 7.** Single site substitution of the 5 N-terminal amino acids of CexE. (A) SignalP result of CexE sequence from ETEC H10407 (E3PPH2_ECOH1) (B) WebLogo of the 5 N-terminal residues of Aap and CexE sequences post Sec signal sequence cleavage. (C) ETEC H10407 *cexE* mutants transformed with pACYC184 (Δ*cexE*) or pACYC-*cexE*-6His with either the wild-type sequence (WT *cexE*) or one of the first five amino acids mutated to glutamic acid. CexE was detected by polyclonal antibodies and RNAP was used as a loading control. The percentage of mCexE was determined from three biological replicates.

The online version of this article includes the following source data and figure supplement(s) for figure 7:

**Source data 1.** ETEC H10407 CexE sequence.
**Source data 2.** Aap and CexE sequences.
**Source data 3.** Aap and CexE sequences with signal sequences removed by SignalP.
**Source data 4.** T-coffee alignment of sequences.
**Source data 5.** Western blots of CexE production.
**Figure supplement 1.** WebLogo of Aap and CexE sequences.
**Figure supplement 2.** Culture supernatant of CexE G20E mutant.
**Figure supplement 2—source data 1.** Coomassie stained gel and western blots of secreted CexE.

MS/MS. A peak was observed in the HLPC trace of mCexE that was not present in uCexE (*Figure 8A*). The mass spectrometry of that peptide identified the five N-terminal residues of CexE with an addition of 238 Da, which is equal to that expected to a single 16 carbon fatty acid addition (*Figure 8B*) confirming the modification of CexE by the addition of an acyl chain onto the N-terminal glycine.

## Heterologous acylation by AatD

Other domains of CexE might be responsible for interacting with AatD other than the N-terminus. Therefore, we fused the N-terminus of mCherry with the signal sequence of CexE and an increasing number of glycine residues to a maximum of three. These constructs were transformed into ETEC H10407Δ*cexE* pCfaD and the production of the mCherry fusions was induced in culture media

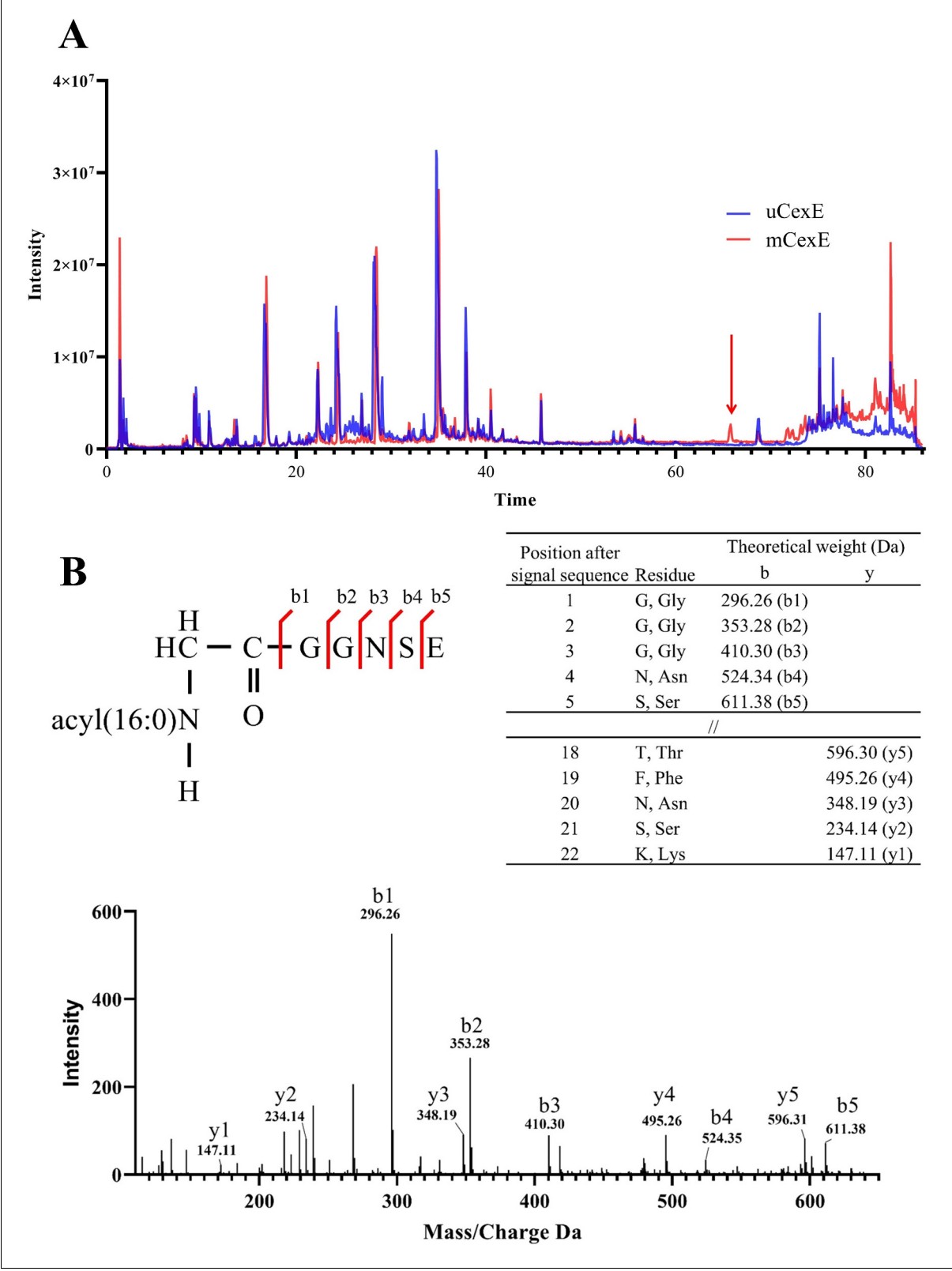

**Figure 8.** Mass spectrometric analysis of modified CexE. (**A**) CexE-6His was isolated from a *cexE* mutant and a *cexE aatD* double mutant and subjected to LC-MS/MS. CexE and pro-CexE were trypsinated and separated by HPLC. (**B**) The indicated peak was subjected to MS/MS to identify the amino acid sequence.

*Figure 8 continued on next page*

*Figure 8 continued*

The online version of this article includes the following source data for figure 8:

**Source data 1.** HLPC of mCexE.
**Source data 2.** HLPC of uCexE.
**Source data 3.** Peptide results.

supplemented with 17-ODYA. The mCherry constructs with N-terminal glycine residues were isolated using the incorporated C-terminal 8 His tag. The purified proteins were subjected to CuAAC with azide-Cy5. The incorporation of 17-ODYA into the mCherry fusions was observed by detecting fluorescence of the Cy5 dye. The signal sequence alone did not cause an incorporation of 17-ODYA (*Figure 9*). However, a single glycine at the N-terminus was sufficient for mCherry acylation, but the addition of further glycine residues to the N-terminus increased the fluorescence signal, indicating an increase of AatD activity proportional with increase in the number of glycine residues present (*Figure 9*). The minimum required for AatD-mediated acylation is a glycine at the N-terminus of the protein post signal sequence. However, an increase in the number of glycine residues appears to increase the efficiency of the reaction.

## Discussion

Post-translational modification of proteins by the covalent attachment of fatty acids occurs for a myriad of proteins in eukaryotic and prokaryotic organisms. Such acylation events confer distinct biochemical properties on the proteins, enabling acylation to regulate intracellular trafficking, subcellular localisation, and molecular interactions. The current dogma for Gram-negative bacterial lipoproteins purports that proteins are triacylated on an N-terminal cysteine through the action of three enzymes Lgt, Lsp, and Lnt (*Grabowicz, 2019*; *Nakayama et al., 2012*; *Zückert, 2014*). While N-palmitoylation of the amino acid glycine does occur in Bacteroidetes, there is no evidence that

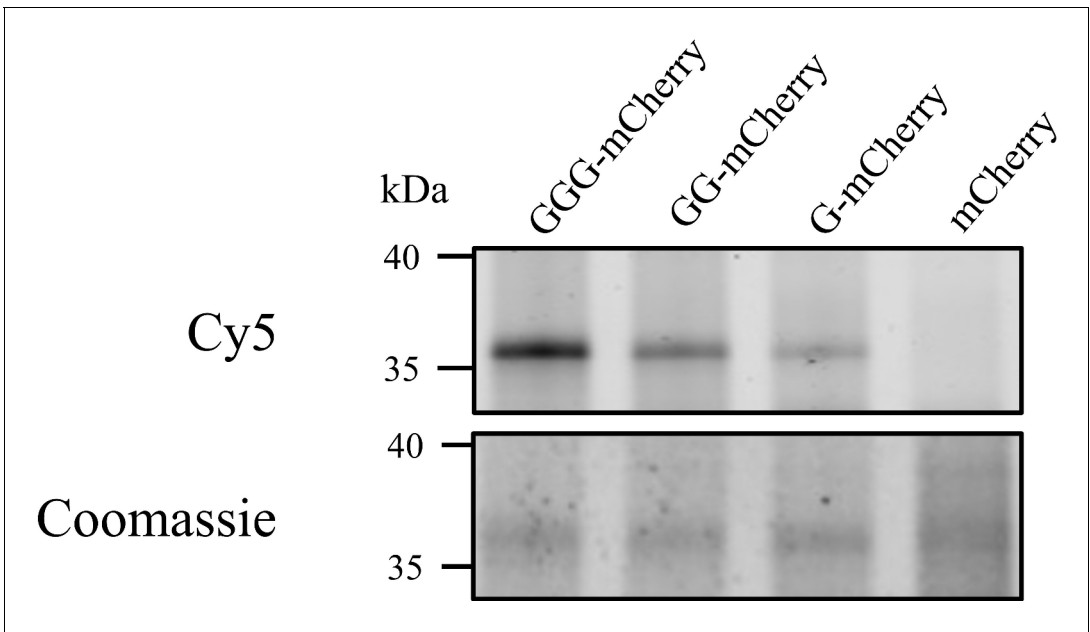

**Figure 9.** Heterologous acylation by AatD. Plasmid encoding mCherry with the CexE signal sequence followed by none, one (G), two (GG), or three (GGG) glycine residues post signal sequence was produced in an ETEC H10407 *cexE* mutant grown in the presence of 17-ODYA. The mCherry proteins were isolated using a C-terminal 8 His tag and azide linked Cy5 was incorporated into mCherry proteins using CuAAC. The acylation of mCherry was detected using fluorescence.

The online version of this article includes the following source data for figure 9:

**Source data 1.** ODYA labelling of chimeric proteins.

this acylated glycine is incorporated into a protein (*Cohen et al., 2015*; *Lynch et al., 2017*). Although N-terminal glycine acylation by *N*-myristoyltransferase is present in life, it is restricted to eukaryotes. N-myristoylation is one of the three major classes of fatty acylation reactions for eukaryotic proteins, which include N-myristoylation of glycine/lysine in proteins such as c-Src or HIV Gag1 (*Resh, 1994*; *Veronese et al., 1988*); S-palmitoylation of cysteine in proteins such as Ras (*Hancock et al., 1989*); and N-palmitoylation of cysteine in proteins such as Hedgehog (*Pepinsky et al., 1998*). Examples of bacterial N-myristoylation rely on eukaryotic proteins produced in *E. coli* or the injection of myristoylatable proteins into eukaryotic cells by type III secretion systems (*Duronio et al., 1990*; *Martin et al., 2011*). While the α-subunit of the heterotrimeric G protein (Gα$_s$) has been reported to be N-palmitoylated at a glycine (*Kleuss and Krause, 2003*), glycine N-palmitoylation is dependent on an adjacent cysteine residue that is known to be S-palmitoylated (*Kleuss and Krause, 2003*; *Linder et al., 1993*). However, subsequent studies suggested that the detected N-palmitoylation of glycine in Gα$_s$ was due to intermolecular transfer of palmitoyl from the S-palmitoylated cysteine to the N-terminal glycine during proteomic analysis (*Ji et al., 2016*). Furthermore, no enzyme was identified as responsible for Gα$_s$ N-palmitoylation. Thus, N-palmitoylation of glycine appears to be a rare or under-described event and to our knowledge this is the first report of enzyme-mediated N-palmitoylation of a protein at a glycine residue. The recent publication by *Belmont-Monroy et al., 2020* identified AatD as an acyltransferase of Aap and CexE responsible for N-palmitoylation of a glycine residue. However, this study failed to recognise this modification as the first confirmed example of enzyme-mediated N-palmitoylation of a glycine or to demonstrate that the position and conservation of the N-terminal amino acids were crucial to acylation. Furthermore, we were able to demonstrate the dependence of CexE secretion on the Sec apparatus that the secreted protein was acylated and that the catalytic triad of the C-N hydrolases is required for AatD-mediated acylation of CexE. Combined with our independently derived results, these data confirm CexE and Aap as the first members of a new class of N-terminally glycine-acylated bacterial lipoproteins.

Since AatD requires the same catalytic residues as Lnt, the two enzymes likely share a similar mechanism of action, albeit with a different substrate protein. Lnt is most efficient at using phosphatidylethanolamine (PE) in vitro but is capable of using the other major phospholipids such as phosphatidylglycerol and phosphatidic acid (*Hillmann et al., 2011*). Although not directly tested in this work, based on similarities with Lnt, PE is likely the major source of palmitoyl for N-palmitoylation of CexE. In fact, *Belmont-Monroy et al., 2020* suggested that PE is the major source of lipid due to an increase of PE in the *aar* mutant compared to the wild type. However, this seems counterintuitive as the deletion of *aar* results in an increase in *aggR* expression, which subsequently results in increased AatD production (*Morin et al., 2013*; *Santiago et al., 2014*; *Yasir et al., 2019*); Aar is a negative regulator of *aggR* expression in EAEC 042 (*Santiago et al., 2014*). If AatD uses PE as a substrate, an increase in the amount of AatD would be expected to lead to a decrease in PE and an increase in lyso-PE, whereas these researchers found a >10-fold increase in PE production and no change in lyso-PE levels in an aar mutant. How lipid homeostasis (*Saha et al., 1996*) is drastically disrupted in such a mutant is intriguing and suggests potential further roles for Aar in phospholipid regulation.

*Belmont-Monroy et al., 2020* suggested that Lnt is able to acylate Aap. However, in our hands there was no evidence of CexE acylation in the absence of AatD. Given that Aap and CexE lack the traditional lipobox of bacterial lipoproteins that would be recognised by the Lol system, that the signal sequence of all Aap and CexE peptides was predicted to be cleaved by signal peptidase I, which has been confirmed experimentally for CexE (*Pilonieta et al., 2007*), that we were able to reconstruct specific AatD-mediated acylation in *E. coli* BL21(DE3), which does not naturally encode AatD or CexE but does encode Lnt, and that modification of CexE could not be detected even when CexE was overexpressed in an AatD-negative Lnt-positive background, we do not believe that Lnt plays a role in CexE acylation. Indeed, our demonstration of acylation of a heterologous protein with a single glycine residue suggests that AatD is a specific *N*-acyltransferase and that it has the potential for exploitation for production of novel acylated peptides in *E. coli*.

The Aat system is, to our knowledge, the only secretion system that acylates and secretes a lipoprotein substrate to the cell surface. While surface exposed lipoproteins have been reported for *E. coli*, for example, Lpp, RcsF, and SslE (*Baldi et al., 2012*; *Cowles et al., 2011*; *Konovalova et al., 2014*), these lipoproteins are acylated by Lgt and Lnt on an N-terminal cysteine in the typical manner and translocated to the outer membrane by the Lol pathway. Similarly, *Neisseria* spp. decorate their

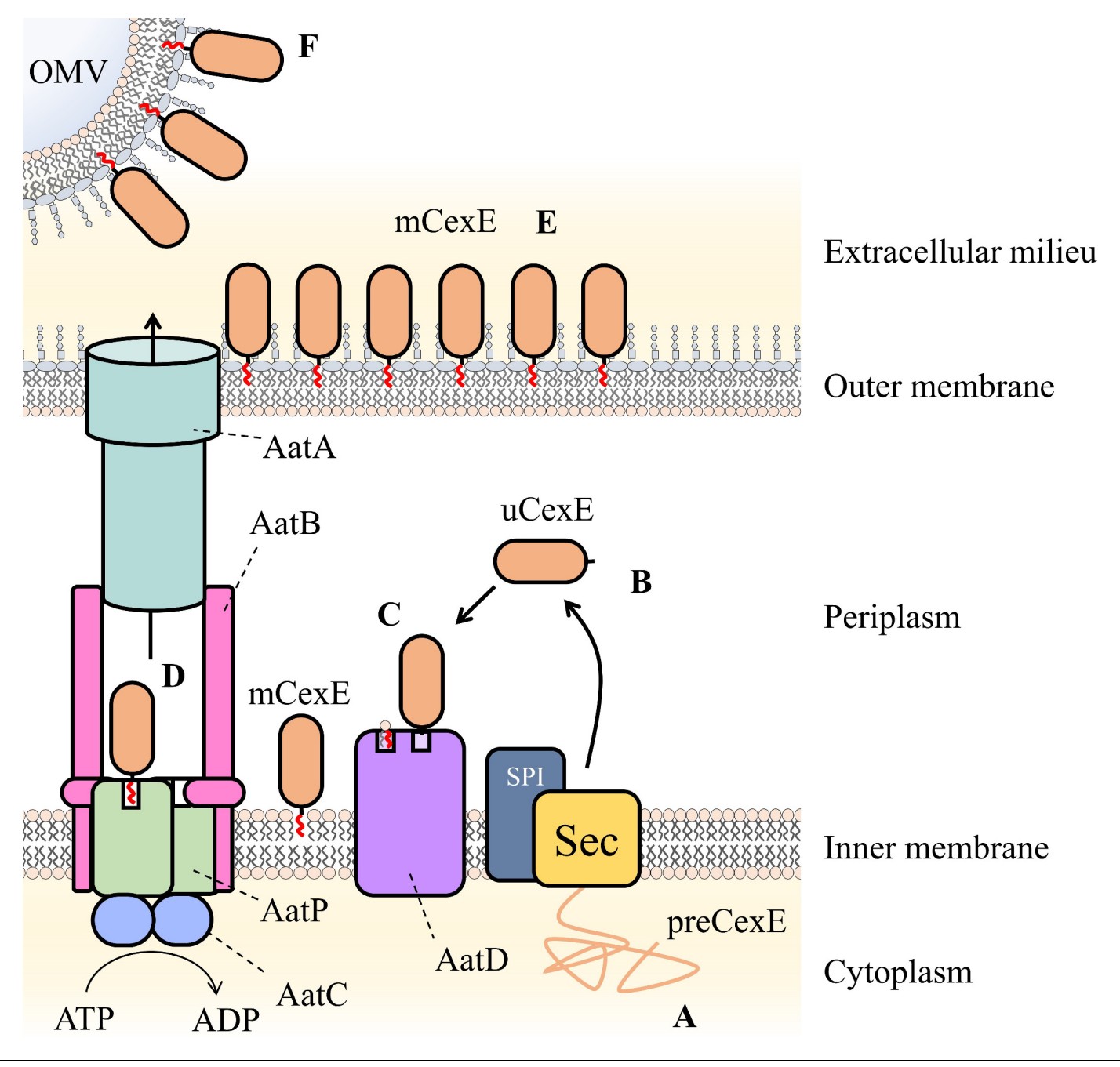

**Figure 10.** Schematic of proposed Aat system mechanism. (**A**) PreCexE produced in the cytoplasm is secreted into the periplasm by the Sec pathway (Sec). (**B**) The signal sequence is cut by signal peptidase I (SPI) resulting in soluble uCexE. (**C**) An acyl chain is added to the N-terminal glycine of uCexE by AatD. Accordingly, mCexE associates with the membrane. (**D**) mCexE is extracted out of the inner membrane by the AatP and AatC complex. A single channel is formed comprised of AatP, AatA, AatB, and AatC that allows the secretion of mCexE. (**E**) mCexE is inserted into the outer leaflet of the outer membrane. mCexE remains associated with the outer membrane by the single acyl chain incorporated onto the N-terminal glycine. (**F**) mCexE is secreted into the extracellular milieu attached to outer membrane vesicles (OMVs).

cell surfaces with secreted lipoproteins that are acylated by Lgt and Lnt on an N-terminal cysteine and trafficked to the outer membrane by the Lol system. In the latter case, an OMP termed Slam is required for translocation of the protein to the cell surface (*Hooda et al., 2016*). In contrast, the Aat system encodes a novel and specialised *N*-acyltransferase, AatD. To allow secretion of the novel lipoprotein CexE, the Aat system has combined proteins of the T1SS and the Lol trafficking pathway.

Not only does the Aat system secrete a novel class of lipoproteins from the periplasm to the surface of the cell but also it acylates that protein itself. For this reason, we believe that reclassification of the Aat system as the first example of the lipoprotein secretion system (LSS) is warranted. We propose that the LSS would remain as a sub-class of the T1SS to emphasise its similarities to the T1SS.

The Aat system is not unique in secreting a periplasmic substrate. Other T1SS employ a two-step secretion mechanism. One such example is the MacAB-TolC complex which secretes heat-stable enterotoxin from the periplasm (*Yamanaka et al., 2008*). We identified MacA and MacB as homologs of AatB and AatPC, respectively, suggesting that AatPC may use a similar mechanism of mechanotransmission as MacB to secrete Aap/CexE (*Crow et al., 2017*). Furthermore, AatB may form a similar gating ring to MacA (*Fitzpatrick et al., 2017*). However, how the Aat system is able to combine the mechanism of the MacAB-TolC T1SS and the LolCDE lipoprotein trafficking pathway to specifically mediate translocation of a lipoprotein to the cell surface has yet to be elucidated.

In *Figure 10*, based on our observations, we propose our model for Aat-mediated protein secretion. First, the Sec machinery secretes preCexE into the periplasm, resulting in cleavage of the signal peptide by signal peptidase I to form uCexE (*Pilonieta et al., 2007*). Once in the periplasm, the N-terminal glycine of uCexE is acylated by AatD to create mCexE. Through this acylation event mCexE then associates with the inner membrane. We believe this to be first step for secretion by the Aat system since CexE acylation by AatD is independent of the other members of the Aat system. In addition, the deletion of any of the *aatPABC* genes does not result in a loss of CexE acylation. We propose that the acylation of CexE is required to enter the AatPABC tunnel. Due to the homology to LolCDE, we believe that mCexE is extracted from the inner membrane by AatPC. Finally, AatA, AatB, and AatPC form a channel for mCexE secretion in a manner analogous to a typical T1SS. Subsequently, mCexE inserts into the outer leaflet of the outer membrane. mCexE remains surface exposed or is further secreted into the extracellular milieu in a soluble form or as a component of outer membrane vesicles, as previously described (*Roy et al., 2011*).

Due to increasing levels of antibiotic resistance in pathogenic organisms, there is a requirement for new protein targets. The conservation of this system within pathogenic bacterial strains represents a possible new drug target. Previous studies have shown that deletion of *aatD*, *aatC*, or *aap/cexE* reduces bacterial colonisation and disease (*Belmont-Monroy et al., 2020*; *Rivas et al., 2020*). Thus, chemicals targeting the Aat system have the potential to prevent or perhaps reduce the severity of bacterial disease. As the Aat system does not appear to be present in non-pathogenic strains, this would be an ideal target to prevent off-target effects. As only pathogenic bacteria would be affected by the inhibition of the Aat system, there would be no risk of evolving resistant alleles in non-pathogenic populations. Furthermore, as the addition of an acyl chain can improve the half-life of peptide drugs like insulin (*Kurtzhals, 2007*), the ability to acylate heterologous proteins with a simple glycine addition is a valuable new tool for the biotechnology enabling new acylated peptides to be produced in *E. coli*. In conclusion, we have identified a novel *N*-acyltransferase. In addition, we have characterised a novel class of bacterial lipoproteins that are acylated and secreted by a composite secretion system. There are still significant questions left to be answered on the mechanism of Aat secretion as well as the function of the secreted proteins.

## Materials and methods

### Key resources table

| Reagent type (species) or resource | Designation | Source or reference | Identifiers | Additional information |
|---|---|---|---|---|
| Strain, strain background (*Escherichia coli*) | H10407 | Evans and Evans (1973) | | Prototypical ETEC strain |
| Strain, strain background (*Escherichia coli*) | 042 | Evans and Evans (1973) | | Prototypical EAEC strain |

*Continued on next page*

*Continued*

| Reagent type (species) or resource | Designation | Source or reference | Identifiers | Additional information |
|---|---|---|---|---|
| Strain, strain background (*Escherichia coli*) | DH5α | New England Bioscience | C2987I | Chemically competent cells |
| Strain, strain background (*Escherichia coli*) | BL21(DE3) | Invitrogen | EC0114 | Protein production strain |
| Recombinant DNA reagent | pKD46 | *Datsenko and Wanner, 2000* | | Plasmid encoding λ Red recombinase genes |
| Recombinant DNA reagent | pCP20 | *Datsenko and Wanner, 2000* | | Plasmid encoding Flp recombinase gene |
| Recombinant DNA reagent | pET26b | Novagen | | T7 expression vector with C-terminal 6 His tag |
| Recombinant DNA reagent | pACYC184 | *Chang and Cohen, 1978* | | Plasmid with p15A origin of replication |
| Recombinant DNA reagent | pACYCDuet-1 | Novagen | | Plasmid with p15A origin of replication and two T7 promoters |
| Antibody | Anti-CexE (rabbit polyclonal) | This paper | | WB (1:2000) |
| Antibody | Anti-His Tag (mouse monoclonal) | GenScript | Cat# A00186, RRID:AB_914704 | WB (1:5000) |
| Antibody | Anti *E. coli* RNA polymerase beta antibody (mouse monoclonal) | BioLegend | Cat# 663903, RRID:AB_2564524 | WB (1:10000) |

## Bacterial cultivation

*Supplementary file 3* contains the bacterial strains used in this study. Bacterial strains were regularly cultivated in LB which consisted of 10 g/l tryptone, 5 g/l yeast extract, and 10 g/l NaCl. Bacterial strains were routinely grown on LB supplemented with 1.25% agar (LBA), and a single colony was used to inoculate liquid cultures for overnight growth at 37℃ with aeration. Overnight cultures were used to inoculate Erlenmeyer flasks containing a fifth of their total volume of LB supplemented with the relevant antibiotics and grown at 37℃ with aeration. Antibiotics were used at the following final concentrations: kanamycin 100 µg/ml, carbenicillin 100 µg/ml, and chloramphenicol 35 µg/ml. For the induction of CexE in ETEC H10407, cultures harbouring pCfaD were grown for 90 min in LB, supplemented with carbenicillin, at 37℃ with aeration. Expression of *cfaD* was induced with 0.2% L-arabinose for 2 hr at 37℃ with aeration. Aap was induced by growing EAEC 042 in DMEM-HG with aeration at 37℃.

## Molecular techniques

Primers for PCR can be found in *Supplementary file 4*. Plasmids used in this study are detailed in *Supplementary file 5*. Unless otherwise stated, all primers were used at a concertation of 10 µM. Q5 High-Fidelity 2X Master Mix (NEB) was used for cloning or mutagenesis. MyTaq Red Mix (Bioline) was used in all other cases. The *cexE* gene from ETEC H10407 was amplified using CcexE-F and CcexE-R primers. Both the vector, pET26b(+) (Novagen), and the insert were cut with NdeI (NEB) and XhoI (NEB) using the CutSmart (NEB) protocol. The digested vector was treated with Antarctic Phosphatase (NEB). T4 DNA Ligase (NEB) was used to ligate the insert DNA to the vector DNA. The ligation mixture was transformed into NEB 5-alpha Competent *E. coli* (High Efficiency). Recovered cells were plated on LBA supplemented with 100 µg/ml kanamycin and incubated at 37℃ overnight. The *cexE* gene and 573 bp upstream of the *cexE* gene, which included the two CfaD binding sites (*Pilonieta et al., 2007*), were amplified by PCR. The reverse primer included a 6His tag. The resulting amplified DNA was digested with SphI and BamHI and ligated into pACYC184. Successful

ligations were selected for as described above on LBA supplemented with 35 µg/ml chloramphenicol. Point mutations were constructed using the QuickChange II method (Aglient). The *aat* genes and *cexE* were disrupted in ETEC H10407 as previously described (*Datsenko and Wanner, 2000*). The plasmid pDOC-K was used as the source of the kanamycin resistance cassette (*Lee et al., 2009*). The *cexE*-mCherry fusion and the *aatD* gene from ETEC H10407 were synthesised by GenScript. Deletions of the *cexE* gene from the plasmid encoding the CexE-mCherry fusion were constructed as previously described (*Moore and Prevelige,, 2002*). Plasmid sequences were confirmed by Sanger sequencing.

## Bioinformatic analysis of the Aat system

The protein sequences of the Aat proteins from ETEC H10407 were used to search the NCBI non-redundant protein sequence database using PSI-BLAST (*Altschul et al., 1997*). Strains encoding a complete Aat system on the same nucleotide accession were used to assess the distance between *aat* genes. Distant homologs of the Aat proteins were identified using HMMER (*Finn et al., 2011*). An HMM was created for each Aat protein using the Aat protein sequences identified by PSI-BLAST. These models were used to search the UniprotKB or Swissprot databases (*UniProt Consortium, 2019*). Protein sequences were aligned using Clustal Omega (*Madeira et al., 2019*). RAxML was used for the construction of phylogenetic trees (*Stamatakis, 2014*). Trees were drawn using iTOL (*Letunic and Bork, 2019*).

## CexE purification for antibody production

BL21(DE3) was transformed with pET26b-*cexE*. An overnight culture of BL21(DE3) pET26b-*cexE* was used to inoculate 2 l of LB supplemented with 100 µg/ml kanamycin. The culture was grown at 37℃ with aeration to an $OD_{600}$ of 0.4. The culture was moved to 20℃ incubator for 30 min prior to induction. CexE production was induced with 50 µM IPTG (Sigma) and incubated overnight at 20℃ with aeration. After overnight growth, cells were harvested by centrifugation at 6000 x *g* for 10 min at 4℃. The cell pellet was resuspended in ice-cold binding buffer (50 mM NaP [77:33 ratio of $Na_2HPO_4$ to $NaH_2PO_4$], 500 mM NaCl, 20 mM imidazole, 0.5 mM TCEP at pH 7.3). Cells were lysed using EmulsiFlex-C3 (Avestin). Cellular debris and intact cells were removed by centrifugation at 10,000 × *g* for 10 min at 4℃. The cellular membrane components were removed by centrifugation at 75,000 × *g* for 1 hr at 10℃. The supernatant was applied to a HisTrap HP 5 ml column (GE Healthcare Life Sciences) overnight at 4℃. The column was washed with five column volumes of binding buffer, then the protein was eluted in 5 ml fractions with elution buffer (50 mM NaP, 500 mM NaCl, 500 mM imidazole, 0.5 mM TCEP at pH 7.3). Samples containing purified protein were concentrated using Vivaspin 6 5000 MWCO columns (Sartorius Stedim). Protein was buffer exchanged using membrane filtration into 50 mM NaP, 250 mM NaCl, and 0.5 mM TCEP at pH 7.3. Purified CexE protein was used to produce primary antibodies by Eurogentec using the 28-day speedy protocol. Polyclonal antibodies against CexE were concentrated prior to use for the detection of CexE.

## Tris-tricine SDS-PAGE and western blotting

Protein samples were separated using Tris-tricine SDS-PAGE as previously described (*Schägger, 2006*). In brief, protein samples were mixed with sample buffer (4% SDS, 2.5% 2-mercaptoethanol, 7.5% glycerol, 0.01% bromophenol blue, 35 mM Tris [pH 7.0]) and separated on 10% Tris-tricine SDS-PAGE gels. Gels were stained with Coomassie brilliant blue or transferred to nitrocellulose for western blotting. Nitrocellulose membranes were covered in 2% BSA (20 g BSA, 2.42 g Tris-base [pH 8.4], 8 g NaCl per litre). The primary antibody was diluted in 2% BSA at the following concentrations: Aap 1 in 5000 (*Sheikh et al., 2002*); CexE 1 in 2000 (this study); and β subunit of RNAP 1 in 10,000 (*E. coli* RNA Polymerase beta Monoclonal Antibody, BioLegend). Primary antibodies were incubated overnight at 4℃ with agitation. Membranes were washed three times in TBST (2.42 g Tris-base, 8 g NaCl, 0.1% Tween-20, pH 8.4 in 1 l) for 5 min at room temperature with agitation prior to incubation with secondary antibody. Anti-rabbit (IRDye 800CW goat anti-rabbit IgG, Li-Cor) or anti-mouse (IRDye 680LT goat anti-mouse IgG, Li-Cor) secondary antibodies were used to detect primary antibody binding. Secondary antibodies were diluted 1 in 15,000 in 2% BSA and incubated for a minimum of 1 hr at room temperature with agitation. Membranes were washed four times with TBST for

5 min at room temperature with agitation. Secondary antibody fluorescence was detected using the Odessy CLx imaging system.

## Inhibition of SecA by sodium azide

Two cultures of ETEC H10407 pCfaD were inoculated from the same overnight culture. The cells were both grown at 37℃ for 90 min with aeration. WCL samples of each culture were taken. L-arabinose was added to both cultures to a final concentration of 0.2%. Sodium azide was added to a final concentration of 3 mM to one of the cultures. Both cultures were grown for 2 hr at 37℃ with aeration. WCL samples of each culture were taken and separated by Tris-tricine SDS-PAGE. CexE was detected by western blotting.

## Proteomic analysis

ETEC H10407 and EAEC 042 were grown in IMDM and DMEM HG, respectively. In initial experiments, CexE and Aap supernatant fractions were harvested with buffer containing Triton X-100 as previously described by *Sheikh et al., 2002*. Subsequently, supernatant proteins from ETEC H10407 pCfaD cultures were isolated using a modification of the protocol described by *Sheikh et al., 2002* where Triton X-100 was excluded, as it proved to be unnecessary. In brief, cells were separated from the culture supernatant by centrifugation at 8000 × $g$ for 10 min at 4℃. The culture supernatant was filtered using Millex-GP Syringe Filter Unit, 0.22 µm, polyethersulfone (Merck), then cooled on ice. Ice-cold 100% TCA was added to a final concentration of 20%. Samples were incubated on ice for 30 min. Proteins were pelleted by centrifugation. The supernatant was removed and discarded from the sample. The pellet was washed twice with 1 ml ice-cold 100% methanol, proteins were collected with centrifugation at 21,000 × $g$ for 15 min at 4℃ between each wash. The supernatant was removed, and residual methanol was evaporated by incubating the sample at 60℃ for 10 min. The pellet was resuspended in 50 µl Tris-tricine sample buffer. If a colour change to yellow occurred, saturated Tris-base was added until the original colour returned. Samples were analysed by Tris-tricine SDS-PAGE.

Membranes were extracted from 50 ml of cells grown under the conditions required for CexE production. Cells were collected by centrifugation and resuspended in 20 ml of 10 mM Tris (pH 8.0), 1 mM phenylmethylsulfonyl fluoride (PMSF). Cells were lysed using EmulsiFlex-C3 (Avestin). Unlysed cells were collected by centrifugation at 5000 × $g$ for 10 min at 4℃. The supernatant was separated. The membranes were collected by centrifugation at 50,000 × $g$ for 60 min at 4℃. The resulting membrane pellet was washed twice with ice-cold 10 mM Tris (pH 8.0). The final pellet was resuspended in 100 µl 10 mM Tris (pH 8.0). The samples were normalised by protein concentration. The membrane samples were separated by Tris-tricine SDS-PAGE.

## CuAAC

Overnight cultures harbouring plasmids encoding His-tagged proteins of interest to be labelled were used to inoculate 25 ml LB supplemented with the relevant antibiotics to an $OD_{600}$ of 0.05. Cultures were grown for 1.5 hr at 37℃. 17-ODYA dissolved in DMSO was added to a final concentration of 20 µM and protein production induced. An equal volume of DMSO was added as a negative control. Cells were collected by centrifugation and resuspended in 600 µl of 50 mM NaP (pH 7.0), 150 mM NaCl, 0.1% Triton X-100, and 1 mM PMSF. Cells were lysed by sonication for 15 min using the bioruptor on a 30 s on, 30 s off cycles. Insoluble material was removed by centrifugation and the supernatant was retained. His-tagged proteins were isolated from the supernatant using Dynabeads His-Tag Isolation and Pulldown (Invitrogen) as per the manufacturer's instructions. Imidazole was removed by dialysis. Protein sample concentrations were measured with Pierce BCA Protein Assay Kit (Thermo Scientific). A master mix was prepared of 0.2 mM Cy5-azide (Sigma-Aldrich) or Alexa Fluor 488 azide (Invitrogen), 0.2 mM Tris[(1-benzyl-1*H*-1,2,3-triazol-4-yl)methyl]amine (TBTA), and 2 mM freshly prepared $CuSO_4$. This was made to 50 µl in $H_2O$ and 5 µl of the master mix was added to 40 µl each normalised protein sample. Sodium ascorbate was added to a final concentration of 5 mM. Samples were incubated at 37℃ for 1 hr with agitation. Protein was precipitated using chloroform-methanol extraction. To each sample, 400 µl MeOH, 150 µl chloroform, and 300 µl $H_2O$ were added and vortexed for 30 s. Samples were centrifuged at max speed for 2 min. The top layer was removed and 400 µl MeOH was added and samples were centrifuged again at max speed for 2 min.

The supernatant was removed, and the pellet was washed twice with 400 µl MeOH, then dried. Pellets were resuspended in 20 µl 1× Tris-tricine sample buffer and separated by Tris-tricine SDS-PAGE. Gels were incubated in fixative (10% acetic acid, 50% methanol) prior to fluorescence detection, then stained with Coomassie brilliant blue.

### Heterologous protein acylation

ETEC H10407 pCfaD was transformed with pRSF-GGG-mCherry, pRSF-GG-mCherry, pRSF-G-mCherry, or pRSF-SS-mCherry. Cultures were grown as previously described for *cfaD* induction. At the same time as *cfaD* was induced, 17-ODYA was added to a final concentration of 20 µM. Cells were grown for 2 hr and harvested by centrifugation. 17-ODYA incorporation was detected as described above.

### Mass spectrometric analysis of CexE acylation

CexE with a 6 His tag was isolated from a cexE mutant and a cexE aatD double mutant of ETEC H10407 pCfaD. The isolated CexE proteins were separated on a Tris-tricine gel and stained with Coomassie brilliant blue. The bands corresponding to uCexE and mCexE were excised and subjected to trypsin-LysC digestion. The tryptic peptide extracts were analysed by nanoHPLC/MS MS/MS on an Eksigent, Ekspert nano LC400 uHPLC (SCIEX, Canada) coupled to a triple TOF 6600 mass spectrometer (SCIEX, Canada) equipped with a PicoView nanoflow (New Objective) ion source; 5 µl of each extract was injected onto a 5 mm × 300 µm, C18 3 µm trap column (SGE, Australia) for 5 min at 10 µl/min. The trapped tryptic peptide extracts were then washed onto the analytical 75 µm × 150 mm ChromXP C18 CL 3 µm column (SCIEX, Canada) at 400 nl/min and a column temperature of 45℃. Solvent A consisted of 0.1% formic acid in water and solvent B contained 0.1% formic acid in acetonitrile. Linear gradients of 2–40% solvent B over 60 min at 400 nl/min flow rate, followed by a steeper gradient from 40% to 90% solvent B in 5 min, then 90% solvent B for 5 min, were used for peptide elution. The gradient was then returned to 2% solvent B for equilibration prior to the next sample injection. The ionspray voltage was set to 2600 V, declustering potential 80 V, curtain gas flow 30, nebuliser gas 1 (GS1) 30, interface heater at 150℃. The mass spectrometer acquired 50 ms full-scan TOF-MS data followed by up to thirty 100 ms full-scan product ion data, with a rolling collision energy, in an information-dependant acquisition mode for protein identification and peptide library production. Full-scan TOF-MS data was acquired over the mass range of 350–1800 and for product ion MS/MS 100–1500. Ions observed in the TOF-MS scan exceeding a threshold of 200 counts and a charge state of +2 to +5 were set to trigger the acquisition of product ion, MS/MS spectra of the resultant 30 most intense ions. The data was acquired and processed using Analyst TF 1.7 software (ABSCIEX, Canada). Protein identification was carried out using Protein Pilot 5.0 for database searching.

## Acknowledgements

This work was supported by the BBSRC DTP scholarship to IRH. We thank Mr Alun Jones for the use of mass spectrometry facilities and help with data analyses. We also thank Dr Tim Wells, Dr Douglas Browning, Dr Matthew Johnson, and Dr James Haycocks for critical advice in development of the project.

## Additional information

### Funding

| Funder | Grant reference number | Author |
|---|---|---|
| Biotechnology and Biological Sciences Research Council | DTP | Adam F Cunningham<br>Ian R Henderson |

The funders had no role in study design, data collection and interpretation, or the decision to submit the work for publication.

## Author contributions

Christopher Icke, Resources, Data curation, Formal analysis, Validation, Investigation, Visualization, Methodology, Writing - original draft, Project administration, Writing - review and editing; Freya J Hodges, Formal analysis, Validation, Investigation, Visualization, Methodology, Writing - review and editing; Karthik Pullela, Resources, Formal analysis, Validation, Investigation, Methodology, Writing - review and editing; Samantha A McKeand, Formal analysis, Investigation, Visualization, Methodology, Writing - review and editing; Jack Alfred Bryant, Resources, Formal analysis, Validation, Investigation, Methodology; Adam F Cunningham, Conceptualization, Supervision, Funding acquisition, Methodology, Project administration, Writing - review and editing; Jeff A Cole, Conceptualization, Supervision, Funding acquisition, Writing - original draft, Project administration, Writing - review and editing; Ian R Henderson, Conceptualization, Data curation, Formal analysis, Supervision, Funding acquisition, Validation, Investigation, Visualization, Writing - original draft, Project administration, Writing - review and editing

## Author ORCIDs

Christopher Icke http://orcid.org/0000-0002-7815-8591
Jack Alfred Bryant http://orcid.org/0000-0002-7912-2144
Ian R Henderson https://orcid.org/0000-0002-9954-4977

## Decision letter and Author response

Decision letter https://doi.org/10.7554/eLife.63762.sa1
Author response https://doi.org/10.7554/eLife.63762.sa2

# Additional files

## Supplementary files

- Supplementary file 1. Nucleotide accession numbers for *aat*-containing bacteria.
- Supplementary file 2. SignalP predictions for CexE proteins.
- Supplementary file 3. Bacterial strains used in this study.
- Supplementary file 4. Primers used in this study.
- Supplementary file 5. Plasmids used in this study.
- Transparent reporting form

## Data availability

All data generated or analysed during this study are included in the manuscript and supporting files.

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
