## [Decision Letter]

**Acceptance summary:**

All three reviewers and both editors were enthusiastic about this paper which describes a hybrid secretion system involved in lipoprotein acylation and trafficking in *E. coli*. The strength and extent of the data, the clever use of genetic, biochemical and bioinformatic approaches, and the high quality writing make this an outstanding study.

**Decision letter after peer review:**

Thank you for submitting your article "Glycine acylation and trafficking of a new class of bacterial lipoprotein by a composite secretion system" for consideration by *eLife*. Your article has been reviewed by three peer reviewers, including Petra Levin as the Reviewing Editor and Reviewer #1, and the evaluation has been overseen by Gisela Storz as the Senior Editor. The following individual involved in review of your submission has agreed to reveal their identity: Syma Khalid (Reviewer #2).

The reviewers have discussed the reviews with one another and the Reviewing Editor has drafted this decision to help you prepare a revised submission.

Summary:

All three reviewers were enthusiastic about the identification and characterization of a hybrid secretion system involved in lipoprotein acylation and trafficking. We were impressed by the strength and extent of the data and the clever use of genetic, biochemical and bioinformatic approaches. At the same time, there was agreement that the conclusion that acylation is involved in CexE secretion is not fully supported. There was also consensus that overlap between this study and Belmont-Monroy et al., 2020 necessitates more direct acknowledgement.

Essential revisions:

1) Additional experiments, potentially using 17 ODYA and the CexE N-terminal substitution mutants, to clarify if acylation is required for secretion.

2) Revisions to introduction and/or conclusion highlighting how this study complements and/or extends the study by Belmont-Monroy.

[Editors' note: further revisions were suggested prior to acceptance, as described below.]

Thank you for submitting your article "Glycine acylation and trafficking of a new class of bacterial lipoprotein by a composite secretion system" for consideration by *eLife*. Your article has been reviewed by Gisela Storz as the Senior Editor, a Reviewing Editor, and two reviewers. The following individual involved in review of your submission has agreed to reveal their identity: Syma Khalid (Reviewer #2).

Summary:

The reviewers appreciated the addition of requested experiment and felt that they improve the quality of the study. At the same time, there was consensus that a few revisions to the text for clarity and completeness need to be made prior to publication. Reviewer 3, in particular, had some straightforward suggestions that should be easy to incorporate. See comments included below for details.

Reviewer #2:

I have nothing to add to my initial review, and my concerns have been more than sufficiently addressed

Reviewer #3:

Discussion.

"To our knowledge, this is the first report of enzyme mediated N-palmitoylation of glycine in nature". Again, even with the glycine qualifier, this statement is not true. First, glycine N-acylation (including palmitoylation) has been described in eukaryotes. In bacteria, N-palmitoylation of glycine has also been defined in Bacteroidetes (and is referenced in the Belmont-Monroy study). This statement doesn't add to the quality, novelty, or relevance of the current work, it should be removed as it is misleading. Changes should be made to "Thus, we describe a new mechanism of N-palmitoylation. We reveal glycine as a new target of N-palmitoylation…"

To help move the field forward, the authors should make clear how they prepare supernatant fractions of "secreted" mCexE. The method used to generate Figure 2 seems to TCA precipitation of filtered supernatants. The method used to generate Figure 4—figure supplement 1 adds a long Triton X-100 treatment prior to TCA precipitation. The method used for Figure 7—figure supplement 2 is not described in any details. Why these methods differ should be explained. L471 describing conditions needs a reference.

Results.

"acylation was required for secretion". The referenced data (Figure 4—figure supplement 1) does not support this statement since the only variable is the presence/absence of lipid-staining dye. The need for acylation is more directly tested in Figure 7—figure supplement 2. This statement should be moved.

Discussion.

"However, this study [Belmont-Monroy et al., 2020] failed to recognise this modification as a previously uncharacterised N-terminal glycine acylation event". This statement is false. An example from that publication refuting the statement: "This peptide is only observed in the 042aap(pAap) sample and corresponds to a protein N-terminal palmitoylation. The post-translational modification of Aap is illustrated in Figure 10E [depicting N-palmitoyl glycine Aap]". Belmont-Monroy et al., make clear that Aap (the CexE homolog) is N-palmitoylated on the glycine liberated after signal peptide processing. That work also showed evidence that phosphoethanolamine is the acyl donor. That finding should also be noted in this manuscript in tandem with their own analysis of functionally important AatD residues.

Concordant findings from separate groups are valuable. The authors' statements should also note how the two studies agree on the nature of N-acylation in these proteins. The authors have clearly provided additional new insights which they highlight well.

Areas of disagreement (e.g. the role of Lnt) are appropriately handled in the text.

Figure 10

The model should include a proposed mechanism for how mCexE is secreted into the supernatant. The authors use this key property in their study to conclude that N-acylation is required for secretion into the supernatant. Readers will need some idea of how mCexE but not uCexE ends up in the supernatant. Outer membrane vesicles (as suggested) might be one way but these also contain periplasmic proteins (such as uCexE) and this should be made clearer.

---

## [Author Response]

Essential revisions:1) Additional experiments, potentially using 17 ODYA and the CexE N-terminal substitution mutants, to clarify if acylation is required for secretion.

We collected the supernatant of EAEC 042 and ETEC H10407 strains grown in the presence of 17-ODYA or DMSO demonstrating that the secreted proteins were acylated. Revision to text “Furthermore, this acylation was required for secretion of Aap and CexE (Figure 4—figure supplement 1).” Additional figure – Figure 4—figure supplement 1.

We collected the supernatant fraction of the G20E SDM mutant demonstrating that alteration of the Glycine abolished acylation and secretion. Revision to text “Moreover, CexE was no longer secreted without the acylation at G20 (Figure 7C—figure supplement 1)”

2) Revisions to introduction and/or conclusion highlighting how this study complements and/or extends the study by Belmont-Monroy.

We added the following wording

“The recent publication by Belmont-Monroy et al., 2020 identified AatD as an acyltransferase of Aap and CexE. However, this study failed to recognise this modification as a previously uncharacterised N-terminal glycine acylation event or to demonstrate that the position and conservation of the N-terminal amino acids were crucial to acylation. Furthermore, we were able to demonstrate the dependence of CexE secretion on the Sec apparatus, that the secreted protein was acylated, and that the catalytic triad of the C-N hydrolases is required for AatD-mediated acylation of CexE. Combined with our independently derived results, these data confirm CexE and Aap as the first members of a new class of N-terminally glycine acylated bacterial lipoproteins.”

“Belmont-Monroy et al., 2020 suggested that Lnt is able to acylate Aap. However, in our hands there was no evidence of CexE acylation in the absence of AatD. Given that Aap and CexE lack the traditional lipobox of bacterial lipoproteins that would be recognised by the Lol system, that the signal sequence of all Aap and CexE peptides were predicted to be cleaved by Signal Peptidase I, which has been confirmed experimentally for CexE (Pilonieta et al., 2007), that we were able to reconstruct specific AatD-mediated acylation in *E. coli* BL21(DE3), which does not naturally encode AatD or CexE but does encode Lnt, and that modification of CexE could not be detected even when CexE was overexpressed in an AatD-negative Lnt-positive background, we do not believe that Lnt plays a role in CexE acylation”

[Editors' note: further revisions were suggested prior to acceptance, as described below.]

Reviewer #3:Discussion."To our knowledge, this is the first report of enzyme mediated N-palmitoylation of glycine in nature". Again, even with the glycine qualifier, this statement is not true. First, glycine N-acylation (including palmitoylation) has been described in eukaryotes. In bacteria, N-palmitoylation of glycine has also been defined in Bacteroidetes (and is referenced in the Belmont-Monroy study). This statement doesn't add to the quality, novelty, or relevance of the current work, it should be removed as it is misleading. Changes should be made to "Thus, we describe a new mechanism of N-palmitoylation. We reveal glycine as a new target of N-palmitoylation…"

We thank the referee for their comments and apologise for not contextualising our work in a clearer fashion in revision 1. We do acknowledge in the discussion that many proteins are acylated, and many are acylated on glycine. However, we are unable to identify from the literature, or databases dedicated to cataloguing such events, any proteins that are clearly N-palmitoylated on glycine by a specific enzyme. In order to clarify the known literature, and how our work relates to previous observations, we have adjusted the introductory paragraph of the Discussion so it reads:

“Post-translational modification of proteins by the covalent attachment of fatty acids occurs for a myriad of proteins in eukaryotic and prokaryotic organisms. Such acylation events confer distinct biochemical properties on the proteins, enabling acylation to regulate intracellular trafficking, subcellular localization, and molecular interactions. […] Furthermore, we were able to demonstrate the dependence of CexE secretion on the Sec apparatus, that the secreted protein was acylated and that the catalytic triad of the C-N hydrolases is required for AatD-mediated acylation of CexE. Combined with our independently derived results, these data confirm CexE and Aap as the first members of a new class of N-terminally glycine acylated bacterial lipoproteins.”

We also adjusted the text:

“We reveal AatD is a new acyltransferase with glycine as the target of N-palmitoylation.”

To help move the field forward, the authors should make clear how they prepare supernatant fractions of "secreted" mCexE. The method used to generate Figure 2 seems to TCA precipitation of filtered supernatants. The method used to generate Figure 4—figure supplement 1 adds a long Triton X-100 treatment prior to TCA precipitation. The method used for Figure 7—figure supplement 2 is not described in any details. Why these methods differ should be explained. L471 describing conditions needs a reference.

We thank the referee for their comment and apologise for any confusion. Our initial experiments used the method previously described by Sheikh et al., where Triton X-100 was used. Subsequent experiments did not use Triton X-100 as we found it was not required. For clarity we have adjusted the description in the methodology. To address this, we have included the following lines:

“ETEC H10407 and EAEC 042 were grown in IMDM and DMEM-HG, respectively. In initial experiments CexE and Aap supernatant fractions were harvested with buffer containing Triton X-100 as previously described by Sheikh et al., 2002. Subsequently, supernatant proteins from ETEC H10407 pCfaD cultures were isolated using a modification of the protocol described by Sheikh et al., 2002 where Triton X-100 was excluded, as it proved to be unnecessary.”

We also added the following wording.

Figure 7—figure supplement 2 legend “Supernatant proteins were isolated by TCA precipitation after removal of cells by centrifugation and filtration.”

Results."acylation was required for secretion". The referenced data (Figure 4—figure supplement 1) does not support this statement since the only variable is the presence/absence of lipid-staining dye. The need for acylation is more directly tested in Figure 7—figure supplement 2. This statement should be moved.

We thank the referee for highlighting this oversight and have added the following wording.

“Furthermore, only the acylated versions of Aap and CexE were present in the culture supernatant (Figure 4—figure supplement 1).”

Discussion."However, this study [Belmont-Monroy et al., 2020] failed to recognise this modification as a previously uncharacterised N-terminal glycine acylation event". This statement is false. An example from that publication refuting the statement: "This peptide is only observed in the 042aap(pAap) sample and corresponds to a protein N-terminal palmitoylation. The post-translational modification of Aap is illustrated in Figure 10E [depicting N-palmitoyl glycine Aap]". Belmont-Monroy et al., make clear that Aap (the CexE homolog) is N-palmitoylated on the glycine liberated after signal peptide processing.

We thank the referee for highlighting this and apologise if our intent was not clear in the first revision. For clarity we have added the following wording.

“The recent publication by Belmont-Monroy et al., 2020 identified AatD as an acyltransferase of Aap and CexE responsible for N-palmitoylation of a glycine residue. However, this study failed to recognise this modification as the first confirmed example of enzyme-mediated N-palmitoylation of a glycine or to demonstrate that the position and conservation of the N-terminal amino acids were crucial to acylation.”

That work also showed evidence that phosphoethanolamine is the acyl donor. That finding should also be noted in this manuscript in tandem with their own analysis of functionally important AatD residues.

We apologise for not addressing this observation earlier. We have now included the following text:

“Since AatD requires the same catalytic residues as Lnt the two enzymes likely share a similar mechanism of action, albeit with a different substrate protein. Lnt is most efficient at using PE in vitro but is capable of using the other major phospholipids such as phosphatidylglycerol and phosphatidic acid (Hillmann et al., 2011). Although not directly tested in this work, based on similarities with Lnt, phosphatidylethanolamine (PE) is likely the major source of palmitoyl for N-palmitoylation of CexE. In fact, Belmont-Monroy et al., 2020 suggested that PE is the major source of lipid due to an increase of PE in the aar mutant compared to the wild-type. However, this seems counter intuitive as the deletion of aar results in an increase in aggR expression, which subsequently results in increased AatD production (Santiago et al., 2016); Aar is a negative regulator of aggR expression in EAEC 042 (Morin et al., 2013; Santiago et al., 2016; Yasir et al., 2019). If AatD uses PE as a substrate an increase in the amount of AatD would be expected to lead to a decrease in PE and an increase in lyso-PE, whereas these researchers found a >10 fold increase in PE production and no change in lyso-PE levels in an aar mutant. How lipid homeostasis is drastically disrupted in such a mutant is intriguing and suggests potential further roles for Aar in phospholipid regulation.”

Concordant findings from separate groups are valuable. The authors' statements should also note how the two studies agree on the nature of N-acylation in these proteins. The authors have clearly provided additional new insights which they highlight well.

We agree and have added the following wording.

“The recent publication by Belmont-Monroy et al., *2020 identified AatD as an acyltransferase of Aap and CexE responsible for N-palmitoylation of a glycine residue.”*

Figure 10The model should include a proposed mechanism for how mCexE is secreted into the supernatant. The authors use this key property in their study to conclude that N-acylation is required for secretion into the supernatant. Readers will need some idea of how mCexE but not uCexE ends up in the supernatant. Outer membrane vesicles (as suggested) might be one way but these also contain periplasmic proteins (such as uCexE) and this should be made clearer.

We thank the referee for pointing out our omission. We have addressed this by altering Figure 10 and the legend to include OMVs and rewording the paragraph discussing Figure 10 as follows. We are unsure of the point the referee is trying to make regarding periplasmic proteins and so have not addressed this further.

“In Figure10, based on our observations, we propose our model for Aat-mediated protein secretion. First, the Sec machinery secretes preCexE into the periplasm, resulting in cleavage of the signal peptide by Signal Peptidase I to form uCexE(Pilonieta et al., 2007). Once in the periplasm the N-terminal glycine of uCexE is acylated by AatD to create mCexE. Through this acylation event mCexE then associates with the inner membrane. We believe this to be first step for secretion by the Aat system since CexE acylation by AatD is independent of the other members of the Aat system. In addition, the deletion of any of the aatPABC genes does not result in a loss of CexE acylation. We propose that the acylation of CexE is required to enter the AatPABC tunnel. Due to the homology to LolCDE, we believe that mCexE is extracted from the inner membrane by AatPC. Finally, AatA, AatB and AatPC form a channel for mCexE secretion in a manner analogous to a typical T1SS. Subsequently, mCexE inserts into the outer leaflet of the outer membrane. mCexE remains surface exposed or is further secreted into the extracellular milieu in a soluble form or as a component of outer membrane vesicles, as previously described (Roy et al., 2011).”